# fMRI predictors based on language models of increasing complexity recover brain left lateralization

**Laurent Bonnasse-Gahot**
Centre d'Analyse et de Mathématique Sociales
CNRS, EHESS
75006 Paris, France
`lbg@ehess.fr`

**Christophe Pallier**
Cognitive Neuroimaging Unit
CNRS, INSERM, CEA, Neurospin center
91191 Gif-sur-Yvette, France
`christophe@pallier.org`

## Abstract

Over the past decade, studies of naturalistic language processing where participants are scanned while listening to continuous text have flourished. Using word embeddings at first, then large language models, researchers have created encoding models to analyze the brain signals. Presenting these models with the same text as the participants allows to identify brain areas where there is a significant correlation between the functional magnetic resonance imaging (fMRI) time series and the ones predicted by the models' artificial neurons. One intriguing finding from these studies is that they have revealed highly symmetric bilateral activation patterns, somewhat at odds with the well-known left lateralization of language processing. Here, we report analyses of an fMRI dataset where we manipulate the complexity of large language models, testing 28 pretrained models from 8 different families, ranging from 124M to 14.2B parameters. First, we observe that the performance of models in predicting brain responses follows a scaling law, where the fit with brain activity increases linearly with the logarithm of the number of parameters of the model (and its performance on natural language processing tasks). Second, although this effect is present in both hemispheres, it is stronger in the left than in the right hemisphere. Specifically, the left-right difference in brain correlation follows a scaling law with the number of parameters. This finding reconciles computational analyses of brain activity using large language models with the classic observation from aphasic patients showing left hemisphere dominance for language.

## 1 Introduction

Since the seminal discovery that language disorders are most often associated with lesions to the brain's left hemisphere (Dax, 1865; Manning and Thomas-Antérion, 2011; Broca, 1865; Wernicke, 1874), the existence of a left-right asymmetry in the cortical processing of language has been amply documented through different approaches, e.g., studies of split-brain patients (Gazzaniga and Sperry, 1967), intracarotid amobarbital injections (Wada and Rasmussen, 1960), electrocortical stimulation (Penfield and Roberts, 1959), functional brain imaging (Binder et al., 1996; Just et al., 1996; Stromswold et al., 1996; Malik-Moraleda et al., 2022), and behavioral measurements such as reaction times to words presented in the left or right visual fields (Hausmann et al., 2019). All in all, even if there is clear evidence that the right hemisphere is implicated in speech and language processing (Bookheimer, 2002; Jung-Beeman, 2005; Lerner et al., 2011; Vigneau et al., 2011; Bradshaw et al., 2017), it is estimated that left hemispheric dominance for language occurs in approximately 90% of healthy individuals (Josse and Tzourio-Mazoyer, 2004; Tzourio-Mazoyer et al., 2017).

38th Conference on Neural Information Processing Systems (NeurIPS 2024).

Given this state of affairs, one can only be surprised by the symmetric patterns highlighted by studies that have relied on computational models of language to predict fMRI brain time-courses in participants listening to naturalistic texts (e.g. Huth et al., 2016; Heer et al., 2017; Toneva and Wehbe, 2019; Caucheteux et al., 2021; Schrimpf et al., 2021; Pasquiou et al., 2023). For example, Huth et al. (2016) constructed word embeddings using a latent-semantic approach based on co-occurrence counts (Landauer and Dumais, 1997) and used them as predictors of brain activity while participants listened to stories. The maps revealed by this approach were strikingly symmetric. This finding was replicated in subsequent studies that have used predictors derived from more advanced language models based on LSTM (Jain and Huth, 2018) or Transformers (e.g. Toneva and Wehbe, 2019; Caucheteux et al., 2021; Schrimpf et al., 2021; Pasquiou et al., 2023) (but see Caucheteux and King, 2022 who reported a significant left-right asymmetry). One potential interpretation is that brain scores are essentially driven by semantic representations (Kauf et al., 2024), supposedly represented in a very distributed fashion across both hemispheres (see e.g. the discussion in Huth et al., 2016).

In this paper, we use 28 large language models (LLMs) of increasing size (from GPT-2 with 124 million parameters to Qwen1.5-14B with 14.2 billion parameters; see Table A.1 for the full list) to fit fMRI data obtained from participants who listened to a naturalistic text (from *Le Petit Prince* dataset, Li et al., 2022). We find that a clear left-right asymmetry emerges as the size and performance of these models increases, and that this is not simply due to differences in signal-to-noise ratio between left and right hemispheres.

## 2 Methods

### 2.1 fMRI data

***Le Petit Prince* dataset.** We use the publicly available fMRI dataset *Le Petit Prince*[1] which provides recordings from English, French and Chinese participants who had listened to the audiobook of *Le Petit Prince* in their native language while being scanned using functional magnetic resonance (TR=2 s; voxel size=$3.75 \times 3.75 \times 3.8$ mm). Technical details on fMRI acquisition and preprocessing can be found in the publication accompanying the dataset (Li et al., 2022). Here, we use fMRI data spatially normalized in the Montreal Neurological Institute space, from all 49 English speakers. All of them were right-handed according to the Edinburgh handedness questionnaire. For each participant, fMRI acquisition was divided into 9 runs lasting each for about 10 min.

**Additional preprocessing and creation of an average subject.** To reduce the computational burden of the study, and because we are interested in making inferences about the general population, we compute a group average from all subjects. In order to further reduce the computational cost of the study, all functional data are first resampled to 4 mm isotropic voxels, close to the original acquisition resolution. Before averaging, we compute a symmetric brain mask common to all subjects using `nilearn compute_multi_epi_mask` function with the threshold parameter set at 50%. The resulting mask, henceforth named *whole brain volume*, comprises 25870 voxels (1656 cm$^3$). For all subjects and each run independently, voxels' time-series are high-pass filtered with a cut-off of 128 s, linearly detrended and standardized (zero mean and unit variance). We then compute the average subject by taking the mean of all these values per voxel and per run. Finally, we trim the first 20 s and the last 20 s of each run, as these were found to present deviation artifacts, and standardize the resulting time-series.

**Inter-subjects reliable voxels.** In order to evaluate the signal-to-noise ratio in each voxel, independently of any language model, we estimate the reliability of each voxel across participants. To do so, we split the group of all 49 subjects into two (almost) equal subgroups (24/25), compute the average response for each group and predict the BOLD time series from each voxel of one group from the activity of all the voxels from the other group. The fitting procedure follows the same method described in more details below when fitting brain response with neural network activations, and is based on a linear mapping from one set of responses to another, evaluated on a held-out run through cross-validation, and regularized using ridge regression (L2). For each run, for each voxel, we thus compute the correlation between the true activity and the activity predicted from the other group of subjects, with a linear model trained on the other runs. This procedure is repeated 10 times, using

---

[1]`https://openneuro.org/datasets/ds003643/versions/2.0.5`

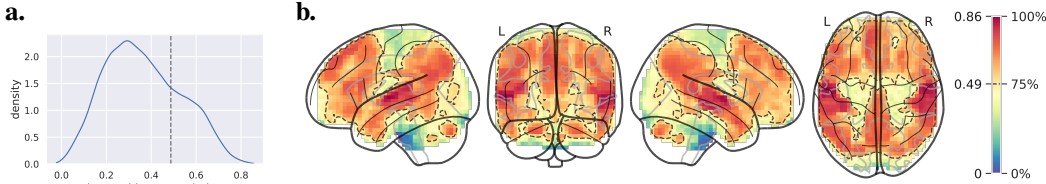

Figure 1: **Inter-subjects reliable voxels**. (a) Distribution of inter-subjects correlations over voxels, computed using two subgroups of subjects and predicting one subgroup average fMRI time-course from the other one (see main text for details). Voxels with a brain correlation above the dotted vertical line represent the 25% voxels with the largest correlations. (b) Glass brain representation of this inter-subjects reliability measure. Hot colors and dotted line show the 25% most reliable voxels.

different splits of the subjects. The final inter-group correlation for each voxel is the average over these 10 trials, and is plotted in Fig. 1. The resulting map is quite consistent to the ones obtained in previous inter-subjects correlations language studies (e.g. Lerner et al., 2011). The plots presented in the main paper are based on the subset of the 25% most reliable voxels, representing a total of 6468 voxels (414 cm$^3$) – 3297 in the left hemisphere and 3171 in the right hemisphere. These voxels are in hot-colored regions delineated with the dashed lines on Fig. 1. In Fig. B.1, inter-subjects correlations averaged over parcels defined by the Harvard-Oxford atlas are plotted. The graphics in panels (b) and (c) reveal a strong relationship between correlations in the left and right homologous regions, with a tendency toward stronger correlations in the left than in the homologous right regions.

**Regions of Interest.** We selected a set of a priori regions of interest from previous works on syntactic and semantic composition: the Temporal Pole (TP), anterior Superior Temporal Sulcus (aSTS) and posterior Superior Temporal Sulcus (pSTS) from Pallier et al. (2011), the Inferior Frontal Gyrus pars opercularis (BA44), triangularis (BA44) and orbitalis (BA47) from Zaccarella et al. (2017), and the Angular Gyrus/Temporal Parietal Junction (AG/TPJ) from Price et al. (2015). Each region was defined as a sphere of 10 mm radius centered on the following coordinates in the Montreal Neurological Institute's MNI152 space : TP (-48, 15, -27), aSTS (-54, -12, -12), pSTS (-51, -39, 3), AG/TPJ (-52, -56, 22), BA44 (-50, 12, 16), BA45 (-52, 28, 10), BA47 (-44,34,-8).

## 2.2 Encoding models

**Language models.** We used 28 pretrained models, all available on the Hugging Face hub, ranging from 124M to 14.2B parameters. The full list along with some details on the number of parameters, layers and hidden size (number of neurons $n_{neurons}$ at the output of each layer) is provided in Table A.1 on page 16 in Appendix. These pretrained models come from 8 different families, namely GPT-2 (Radford et al., 2019), OPT (Zhang et al., 2022), Llama 2 (Touvron et al., 2023), Qwen (Bai et al., 2023), gemma (Team et al., 2024), Stable LM (Bellagente et al., 2024), Mistral (Jiang et al., 2023), and Mamba (Gu and Dao, 2023). In this study, we only consider base models, trained with the same next-word prediction task. Most models are based on the Transformer decoder architecture (Vaswani et al., 2017), apart from the Mamba family, which is a recent come-back of the recurrent neural network approach that is competitive with the Transformer language models.

In order to extract the activation of each model in response to the text of *The Little Prince*, each model is fed with the full original English text. We make use of the time-aligned speech segmentation provided in the *Le Petit Prince* openneuro.org repository, to align the activity of the neural networks with what the subjects heard in the scanner. Each word in the text is associated with its onset in the audiobook, and the representation that we compute is the average of all the vectors of all the tokens that constitute this word, as well as the following punctuation marks if there are any. Finally, in order to mimic the BOLD response, the activity of the artificial neural network under consideration is convolved with the Glover haemodynamic response function (Glover, 1999), as implemented in the `nilearn` package.

**Baseline models.** In addition to considering all these pretrained language models, we look at a few baselines to compare with. First, we consider purely random vectors, aligned with the word onsets. Second, we consider random embeddings, similar to the previous case, but where each word is always associated with the same (random) vector. In these two cases, we look at 300 and 1024 dimensional

vectors. The results presented in this paper for these random baselines were averaged over 10 models obtained from different seeds. Third, we look at non-contextual embeddings provided by the seminal GloVe embeddings, where each word is always associated with the same vector, of 300 dimensions here, after training on co-occurrences on large English text corpora (Pennington et al., 2014).

**Procedure for fitting an encoding model.** For each model, we separately consider the activity provided by each layer, plus the embedding layer. Hence, for a 12-layer model, we consider 13 layers. Let us notate $X_{l,k}$ the activity of a layer $l$ generated by the text of run $k$, (after convolution with the haemodynamic response function, as described above), and $Y_{v,k}$ the BOLD time series of voxel $v$ of run $k$. $X_{l,k}$ is a $n_{\text{scans},k} \times n_{\text{neurons}}$ matrix, and $Y_{v,k}$ a vector of dimension $n_{\text{scans},k}$, where $n_{\text{scans},k}$ is the number of scans of run $k$, and $n_{\text{neurons}}$ the number of dimensions of layer $l$ (for instance, gpt2-large model has 1280 dimensions, and Mistral-7B-v0.1 has 4096 dimensions). The goal is to predict the functional brain activity $Y_{v,k}$ using the artificial neural activity $X_{l,k}$ as regressors, simply using a linear mapping between the two: $\widehat{Y_{v,k}} = X_{l,k}\beta$, where $\beta$ is a vector of dimension $n_{\text{neurons}}$. In order to avoid overfitting, the $\beta$ coefficients are determined using ridge regression with regularization strength controlled by the parameter $\alpha$. More precisely, given a run $k$ used as test, all the remaining 8 runs are used as training set to determine the $\beta$ coefficients. The hyperparameter $\alpha$ is chosen using nested cross-validation among a range of possible values (16 values log-spaced between $10^2$ and $10^7$): among the runs used for training, one is used for validation, and the remaining 7 runs for training the ridge regression with each value of $\alpha$. Correlation between the actual values $Y_{v,\text{val}}$ and the predicted ones $\widehat{Y_{v,\text{val}}}$ is used to decide which value of $\alpha$ is used during training: the best $\alpha$ is chosen as the one yielding the greatest correlation averaged over all voxels [2]. This value is then used to compute the ridge regression on the training runs and predict the functional brain activity on the test run $k$. The brain correlation for a given voxel $v$ and a given layer $l$ is given by repeating this procedure for all 9 runs and taking the average correlation: $1/n_{\text{runs}} \sum_{k=1}^{n_{\text{runs}}} \text{corr}(Y_{v,k}; \widehat{Y_{v,k}})$. In the end, for a given voxel, we take as brain correlation the best correlation when considering all the layers of the model.

## 2.3 Computer code

The Python 3.10 code written for the present project relies on the following libraries: `transformers` v4.40.1 (Wolf et al., 2020), `scikit_learn` v1.4.2 (Pedregosa et al., 2011), `nilearn` v0.10.4, `Pytorch` v2.3.0 (Paszke et al., 2019), `matplotlib` v3.8.4 (Hunter, 2007), `seaborn` v0.13.2 (Waskom, 2021), `numpy` v1.26.4 (Van Der Walt et al., 2011), `pandas` v2.2.2 (McKinney et al., 2010), `statsmodels` v0.14.2 (Seabold and Perktold, 2010). All pretrained models were downloaded from Hugging Face through the `transformers` interface. The code is available at `https://github.com/l-bg/llms_brain_lateralization`.

The main English study on the average subject takes about five days of CPU time on a computer equipped with an Intel Xeon w5-3425 processor (12 cores), mainly dedicated to perform the ridge regression of all layers of all models, using the `scikit-learn` package.

## 3 Results

### 3.1 Distribution of correlations

The distributions of correlations over voxels, for each model, are displayed on panel (a) of Fig. 2 for the 25% most reliable voxels, and of Fig. B.2 for the whole brain volume. The random vector baseline yields very low brain correlations centered around 0. Random embeddings (fixed random vector for each word) show positive correlations, especially in the 25% most reliable voxels mask, and it increases with the dimension of the embeddings from 300 to 1024. GloVe word embeddings perform better than these baselines. Finally, the contextual embeddings provided by the large language models yield even higher brain correlations.

---

[2]We performed exploratory analyses showing that a single value for $\alpha$ for all the voxels in the brain was not detrimental to the predictions of the functional data, and produced a massive gain in speed.

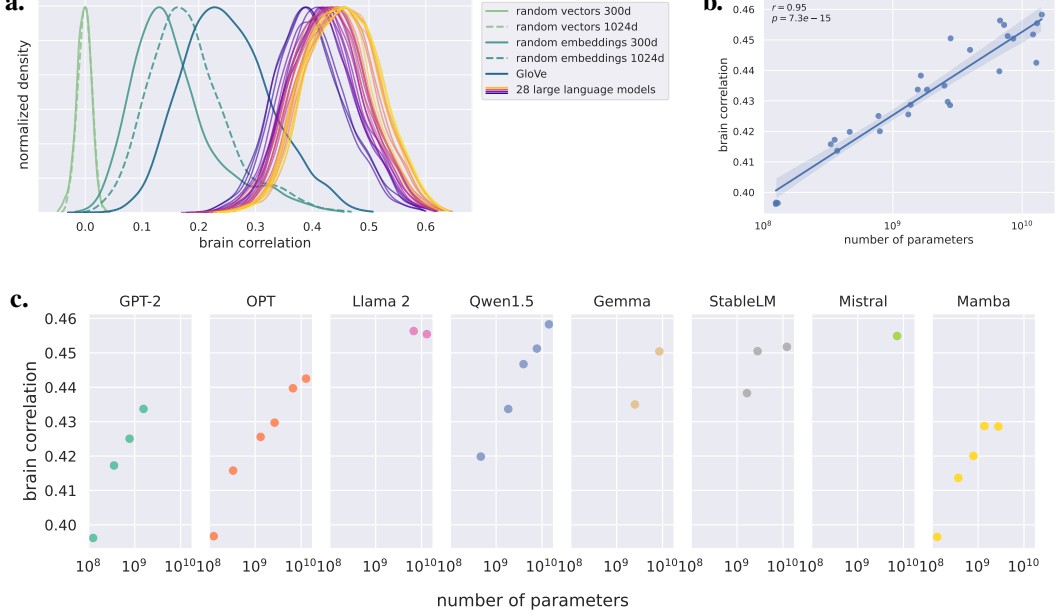

Figure 2: **Performance of various models in predicting fMRI brain time-courses**. (a) Density estimates of the distributions of $r$-scores obtained for all 28 large language models, the random baselines and GloVe. The densities are scaled to have the same maximum (b) Average $r$-score as a function of the number of parameters of the model, in log scale. Here and in the next figures, the shaded area indicates the 95% confidence interval of the slope, computed with bootstrap. (c) Same, split by models' family.

## 3.2   Scaling law of fMRI encoding models

Focusing on the contextual large language models, Fig. 2b shows the relationship between the size of the language models and their prediction performance, that is, the average correlation across the 25% most reliable voxels. The linear correlation between the brain correlation and the logarithm of the number of parameters is $r = 0.95, p = 7.3e - 15$. The corresponding plot for the whole volume, displayed in panel (b) of Fig. B.2, also shows a linear relationship ($r = 0.95, p = 3.2e - 14$). This finding replicates the scaling law first described by Antonello et al. (2024).

Fig. 2c details the results by the family of models. Older models like the ones from the OPT family perform worse than more recent ones like Mistral or Qwen, although within the same family the scaling law holds well in general, as exemplified by the GPT-2, OPT or Qwen1.5 families. Interestingly, the Mamba family, which is based on a recurrent neural network architecture contrary to all the other, Transformer-based, models, has similar brain scores, although they lie at the bottom of the envelope of the results (see Fig. B.3). In addition, we performed an analysis of the fit as a function of layer depth, depicted in Fig. B.4. The results confirm earlier reports by Toneva and Wehbe (2019) and Caucheteux and King (2022) that the middle layers are the most predictive ones.

**Brain maps of smallest vs. largest models.**   Fig. 3 shows the brain correlations maps associated with the two most extreme models: GPT-2 with the smallest number of parameters and Qwen1.5-14B with the largest number of parameters. Interestingly, while the map generated by GPT-2 is quite symmetric, Qwen1.5-14B fits fMRI responses better in the left hemisphere than in the right (while overall performing better than GPT-2).

**Voxel-wise sensitivity to model size.**   To investigate the strength of the relationship between model size and brain score in each voxel, we compute the slope of the linear regression line between the logarithm of the number of parameters and the correlation score. These slopes are displayed in Fig. 4. The strongest effects of model size are detected in the left angular gyrus, the medial prefrontal cortex and the precuneus on the medial surface of the parietal lobe. Smaller effects are detected in the middle temporal and inferior frontal gyri.

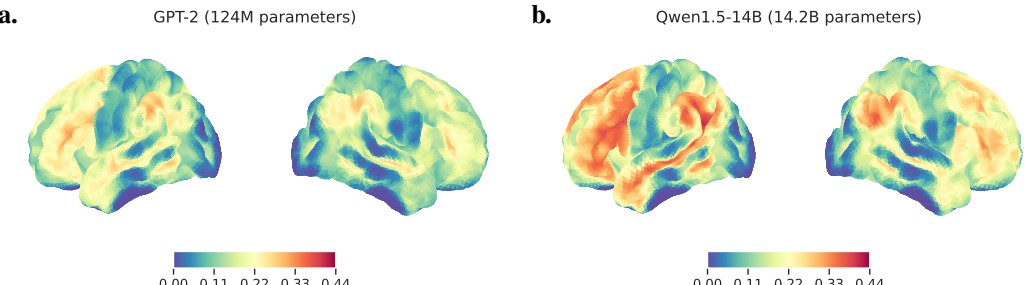

Figure 3: **Brain correlation maps associated with the smallest (a) and the largest model (b).** These maps show the increase in $r$-score relative to the model using the random embedding baseline 1024d.

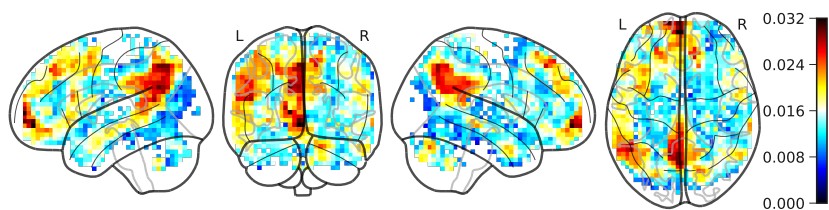

Figure 4: **Voxel-wise strength of the relationship between models' size and their predictive power.** The slopes of the linear regression between $r$-score and the logarithm of the number of parameters are presented on a glass brain view. For readability, only voxels with $p$-values smaller than $10^{-7}$ are shown.

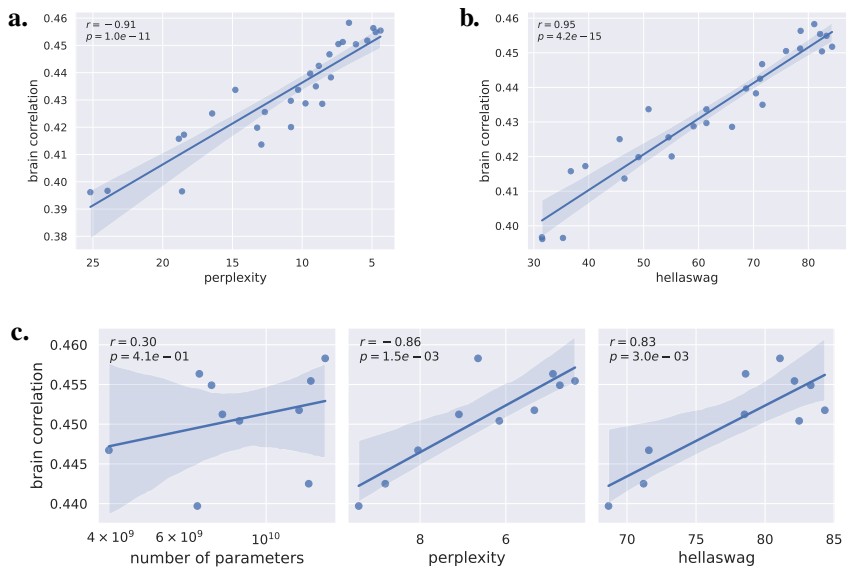

Figure 5: **Brain correlation and performance on natural language tasks.** (a) Brain correlation on the 25% most reliable voxels for all 28 large language models as of a function of perplexity on Wikitext-2 test set. Note that the x-axis is inverted, as the lower the perplexity the better the model. (b) Same with performance on the Hellaswag benchmark. The higher the better. (c) Same as Fig. 2b and (a) and (b), but focusing on the 10 largest models, with a number of parameters above 3B. See Fig. B.5 for similar plots on the whole brain volume.

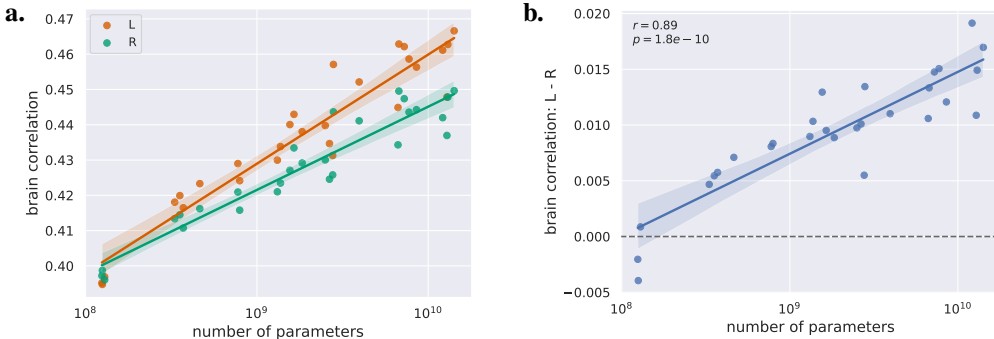

Figure 6: **Emergence of left lateralization with the size of the encoding models**. (a) Brain correlations on the 25% most reliable voxels as a function of the number of parameters, for the left hemisphere (red) and for the right hemisphere (green). (b) The difference between left and right hemispheric brain correlations also shows a scaling law.

**Brain correlation and performance on natural language tasks.** The number of parameters of a model is a rough proxy of its performance on language tasks. For example, OPT-13B is found to have lower performances on various natural language processing tasks than Mistral-7B which has almost half the number of parameters. Hence, we also look at measures of models' performance beyond their raw number of parameters. First we compute the perplexity of all the models on the Wikitext-2 test set (Merity et al., 2016). Second we look at the Hellaswag benchmark (Zellers et al., 2019), which aims to measure the ability of a model to understand language. This measure was found to be quite challenging for language models, and the performance of models scale on this benchmark well with the training budget, either in terms of number of parameters or number of tokens seen during training (see Touvron et al., 2023). Fig. 5 shows how brain correlation increases with better models, either as measured by lower perplexity on the Wikitext-2 test set or on the Hellaswag benchmark. Both measures exhibit strong correlation with brain fit ($r = -0.91, p = 1.1e - 11$ for perplexity and $r = 0.95, p = 4.2e - 15$ for Hellaswag). But this is particularly relevant when looking at the ten largest models, with number of parameters above 3 billions: whereas the correlation between brain score and number of parameters is no longer significant in this case ($r = 0.30, p = 0.41$), the other measures of performance display a significant correlation ($r = -0.86, p = 1.5e - 3$ for perplexity and $r = 0.83, p = 3.0e - 3$ for Hellaswag): see Fig. 5.

### 3.3 Left-right hemispheric asymmetry

**Voxel-based analysis.** The maps presented on Fig. 3 and Fig. 4 hint at left-right differences. Therefore, we plot the relationships between model size and brain scores split by hemisphere in Fig. 6a. This graphic reveals that the larger the model, the stronger the left-right asymmetry. More precisely, while the smallest models have similar $r$-scores both in the left and right hemisphere, the largest models yield stronger $r$-scores in the left one. The interaction between model size and hemisphere can be assessed by the correlation between model size and the left-right difference in $r$-scores, displayed in Fig. 6b. This graph shows that the difference also follows a scaling law ($r = 0.89, p = 1.8e^{-10}$). See Fig. B.6 for a similar analysis on the whole brain volume. Finally, as we did for the brain scores (Fig. B.4), we examined the effect of layer depth on the asymmetry. This analysis, reported on Fig. B.7, shows that the layers with the stronger fit also have the strongest left-right asymmetry.

**Regions of Interest analysis.** The relationships between model size and $r$-scores in seven regions of interest from the language network in the left hemisphere, and in their mirror-image regions in the right hemisphere, are shown on Fig. 7. Brain scores improve with model size in all regions, both in the left and in the right hemisphere. Except for pars opercularis (BA44) in the inferior frontal gyrus, and for the temporal pole (TP), a left-right asymmetry in favor of the left emerges when model size increases. The steepest effect occurs in the Angular gyrus/Temporo-parietal. This region, AG_TPJ, was already highlighted in Fig. 4 showing the slopes in individual voxels.

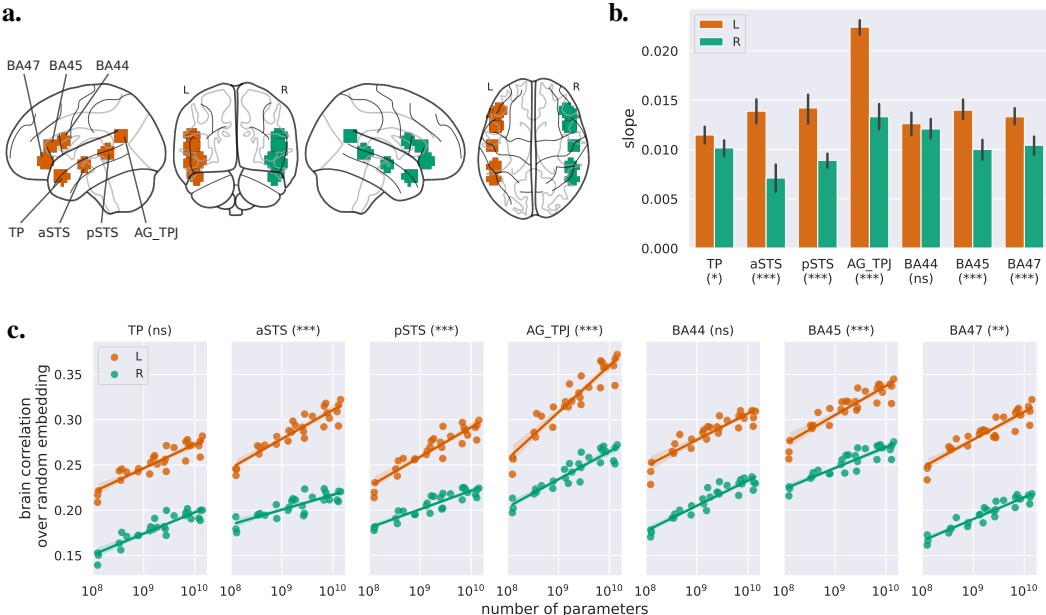

Figure 7: **Model size impact in various Regions of Interest**. (a) Locations of ROIs. (b) Slope in $r$-score as a function of the logarithm of the number of parameters, averaged over all voxels of a given ROI. Error bars indicate 95% confidence interval, estimated by bootstrap. The annotation below each ROI label indicates an estimate of the level of statistical significance of the difference between the slopes of the voxels of a left ROI compared to its respective right ROI (ns: non significant, *:$p < 0.05$, **:$p < 0.01$, ***:$p < 0.001$). (c) Brain correlations over the random embedding baseline (see Fig. B.9 on page 22 for similar plots with raw brain correlations), as a function of the logarithm of the number of parameters. The annotation next to each ROI label indicates the level of statistical significance of the correlation between the difference in $r$-scores of the models on the left hemisphere minus the right hemisphere, as a function of the logarithm of the number of parameters, testing the interaction between hemisphere and model size. See Table A.2 for full statistics.

**Relationship between signal-to-noise ratio, brain scores, and growth in asymmetry.** Do large models fit the left hemisphere better than the right simply because the signal-to-noise ratio, or ceiling of explainable signal, is higher in the left voxels than in the right ones? A proxy for the explainable signal is the model-free inter-subject correlation (ISC). Fig. B.1 shows that the ISC indeed tends to be stronger on the left. To determine if this could explain our findings, we conducted several analyses presented on Fig. B.8 and Fig. B.9. First, we computed "normalized $r$-scores" by dividing raw $r$-scores by the ISC in each voxel. Panels (a) and (b) of Fig. B.8 display the relationship between these normalized brain correlations and model size. The left-right asymmetry previously observed with raw $r$-scores on Fig. 6 still holds. Next, using the parcels from the Harvard-Oxford atlas, we plot the relationship between ISC, brain correlations and their asymmetry (Fig. B.8, bottom panels). Although ISC and brain correlation correlate (panel c), the left-right difference in slope of brain correlation as a function of model size does not covary with the interhemispheric difference in ISC (panel d). Finally, we computed the ISC in our Regions of Interest. Fig. B.9 suggests that while differences in ISC partly explain the average brain score difference, they do not explain the increase in difference between left and right hemispheres, across models.

**Impact of model training.** In order to assess the impact of model training on the left-right asymmetry, we compared $r$-scores for randomly initialized, untrained models, and for trained models, from the GPT-2 and the Qwen1.5 families. Fig. B.10a shows that, with training, brain correlations get better on the left hemisphere compared to the right one. Moreover, while the left-right score difference increases with the number of parameters for trained models, this relationship breaks down and is flat in the case of the untrained models. Another way to assess the influence of model training is to fit the same LLM at various stages of training. The Pythia family (Biderman et al., 2023) provides many intermediate checkpoints during training, from the untrained model, randomly initialized,

to the full model, trained on one epoch on the training set of 300B tokens from the Pile dataset. Here we consider Pythia-6.9B, with 32 layers and 4096 dimensions. Fig. B.10b shows the impact of training on brain correlations. The model's fit improves as training proceeds and the left-right asymmetry, originality biased towards the right hemisphere, increases, reaching a plateau favoring the left hemisphere.

### 3.4 Extension to other languages

As left dominance is universal (see e.g. Malik-Moraleda et al., 2022; it is even attested in sign language, Poizner et al., 1987), we wanted to check if our result holds for other languages. We therefore applied our approach to Chinese and French data provided in *Le Little Prince* dataset (considering all 35 Chinese participants and all 28 French participants).

We first computed inter-subjects correlations. Their distributions and the corresponding maps are displayed on Fig.B.11 and Fig.B.12 panels (a) and (b). The correlation values are lower than for English (Fig.B.1), but the most reliable voxels encompass the same brain regions. Panels (d) and (e) reporting correlations in regions from the Harvard-Oxford atlas (shown in panel (c)) and their left-right difference qualitatively replicate the previous observations in English results, that is, left-right asymmetries are detected in the same regions.

Although much fewer LLMs are available than for English, some models on the Hugging Face hub were trained on Chinese or French data. We considered 10 models for Chinese, ranging from 12M to 14B parameters, and 4 models for French, ranging from 124M to 7B parameters (see panel (f)). The relationships between model size and $r$-scores are presented in panels (g), (h) and (i) of Fig. B.11 and Fig. B.12. They confirm the scaling law, and the emergence of the left-right hemispheric asymmetry in two languages different from English.

### 3.5 Individual analyses

The analyses presented above were performed on data averaged across individuals. It is important to verify if the emergence of asymmetry holds at the individual level or is an artifact of averaging. Because of limitations in time and computational power available to us, we could only analyze five participants on a subset of models. To avoid any bias, we selected the first 5 English participants. The results, presented in Fig. B.13, show that three of them present a significant increase in asymmetry ($p < 0.05$) when considering the full brain (panel c), and four when considering the 10% voxels with the highest $r$-scores in each hemisphere (panel e).

## 4 Discussion & Conclusion

By manipulating the size of artificial neural language models used to fit fMRI data of naturalistic language comprehension, we observed: (1) a linear relationship between the logarithm of the number of parameters of models and their ability to predict the fMRI time-courses (Fig. 2) and (2) a left-right asymmetry emerging when encoding models are based on increasingly complex models (Fig. 6).

The first observation replicates the finding of Antonello et al. (2024) who described a scaling law between model size and brain score, extending to brain data the scaling laws observed between model size and performance on various NLP tasks (Kaplan et al., 2020). Here, we generalize their result to a different dataset that includes three languages, and using a larger variety of models and families that notably include a non-Transformer, Mamba.

The scaling law breaks down when focusing on the largest models, above 3B parameters. Yet, in this range, indices assessing the performance of the model on NLP tasks are still predictive of brain score (Fig. 5). The relationship between neural network models performance in NLP tasks and brain scores, was also investigated by Schrimpf et al. (2021) and by Caucheteux and King (2022). Both teams reported that brain scores generally improved as the performance in the next word prediction task increased, extending to the field of language processing results that were found in the visual domain by Schrimpf et al. (2020) or in the auditory domain by Kell et al. (2018). Caucheteux and King (2022) found that brain scores reached a plateau, and even dropped somewhat (see their Fig.2g) which might be due to an overspecialization for the next word prediction task in their study (Caucheteux et al., 2023).

Our second observation is the emergence of an asymmetry with increasing model size, a finding that has not been reported in the literature [3]. A quick but vague explanation is that larger models better capture language representations in the left hemisphere. It begs the question of which representations are improved in larger models relative to smaller ones. For example, is it lexical, syntactic, semantic, pragmatic or world knowledge? The present work calls for detailed comparisons of the features discovered by large vs small language models, and how and where this translates into better fMRI prediction.

One can attempt to perform inverse inference, that is, try to infer which processes are involved from the brain areas highlighted by our analyses, keeping in mind the pitfalls of such an approach (Poldrack, 2011). The cortical areas where model size has the strongest impact are the Angular gyrus, the Precuneus and the medial prefrontal cortex (mPFC). These regions are also those where the effects of training a (small) network are the strongest Pasquiou et al. (2022) and were fit by models trained on semantic but not on syntax by Pasquiou et al. (2023). These regions are not specific to language, e.g. they are part of the default mode network, but are involved in the highest levels of language comprehension (Simony et al., 2016; Chang et al., 2022). The mPFC and Precuneus are known to be sensitive to discourse coherence (Ferstl and von Cramon, 2001; Xu et al., 2005). The Angular Gyrus is also considered part of the semantic system (Seghier et al., 2010; Binder and Desai, 2011; Price et al., 2015; Kuhnke et al., 2023). This suggests that the main effect of increasing model size is to improve the model capabilities at the semantic and pragmatic levels. The analysis by regions of interest revealed that the asymmetry holds in most regions of the core language network, except pars opercularis (BA44) and the Temporal Pole. The left BA44 has been associated with syntax (Zaccarella and Friederici, 2016) and/or speech processing (Matchin and Hickok, 2020). The temporal poles, both on the left and on the right, are linked to semantic processing (Pobric et al., 2010; Pylkkänen, 2019) and have been designated as an amodal semantic hub by Patterson and Lambon Ralph (2016) (but see Snowden et al., 2018).

It is remarkable that in all regions of interest, even in the right hemisphere, the largest models' fits always improve over the smallest models (see Fig.7). It has been known for long that the right hemisphere has some language capacity (Bradshaw et al., 2017), which can manifest in split brain patients or in patients with lesions in the left hemisphere (Vigneau et al., 2011). The right hemisphere has been associated with speech processing, especially prosody, usually considered to be processed in the right temporal lobe (Wildgruber et al., 2006), as well as with high-level aspects of language understanding involved in the comprehension of metaphors or jokes (Jung-Beeman, 2005; Bookheimer, 2002). More work will be needed to determine which aspects of LLMs allow them to fit these regions.

Our main analyses were performed on fMRI data averaged across participants because of computational power limitations. We report preliminary analyses on 5 participants showing that the emergence of asymmetry holds at the individual level (see Fig. B.13). Nevertheless there is an important variability between subjects in brain correlations, in the difference between left hemispheric vs. right hemispheric average brain correlations, and in the slope of the left-right difference. It would be interesting for future work to assess the reliability of these measures, and to compare them with independent assessments of individual language hemispheric dominance. Unfortunately, such measurements are not provided in the fMRI dataset that we use.

An interesting question concerns the evolution of lateralization during the training process of large language models, either by training such a model from scratch or by having access to checkpoints during the training (see Antonello et al., 2024; Pasquiou et al., 2022, for first attempts). In humans, it seems that language processing in the brain starts bilaterally and then becomes progressively lateralized over time (Szaflarski et al., 2006; Olulade et al., 2020; Ozernov-Palchik et al., 2024), although some studies have found very early lateralization (Witelson and Pallie, 1973; Wood et al., 2004). Our preliminary exploration with the Pythia-6.9B model (Fig. B.10) reveals that the left-right asymmetry emerges during the training process. This leaves open the questions of which representations improved by learning are responsible for this asymmetry, and whether parallels with the development of language acquisition can be drawn.

---

[3]Note that the growing asymmetry with model size goes beyond showing a left-right difference in a single model, which could be explained by signal-to-noise ratio asymmetry (see, e.g., Caucheteux and King, 2022, Fig. 2d)

# 5 Acknowledgments

We would like to sincerely thank all four anonymous reviewers, whose constructive feedback significantly contributed to the improvement of the paper. Many thanks also to Yair Lakretz, Emmanuel Chemla, and all the participants of the Linguae ML seminar at École Normale Supérieure.

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

# A   Supplementary Tables

Table A.1: List of the 28 English large language models included in the study, grouped by family.

| model name | $n_{\text{parameters}}$ | $n_{\text{layers}}$ | $n_{\text{neurons}}$ |
|---|---|---|---|
| gpt2 | 124M | 12 | 768 |
| gpt2-medium | 355M | 24 | 1024 |
| gpt2-large | 774M | 36 | 1280 |
| gpt2-xl | 1.56B | 48 | 1600 |
| opt-125m | 125M | 12 | 768 |
| opt-350m | 331M | 24 | 1024 |
| opt-1.3b | 1.32B | 24 | 2048 |
| opt-2.7b | 2.65B | 32 | 2560 |
| opt-6.7b | 6.66B | 32 | 4096 |
| opt-13b | 12.9B | 40 | 5120 |
| Llama-2-7b-hf | 6.74B | 32 | 4096 |
| Llama-2-13b-hf | 13.02B | 40 | 5120 |
| Qwen1.5-0.5B | 464M | 24 | 1024 |
| Qwen1.5-1.8B | 1.84B | 24 | 2048 |
| Qwen1.5-4B | 3.95B | 40 | 2560 |
| Qwen1.5-7B | 7.72B | 32 | 4096 |
| Qwen1.5-14B | 14.17B | 40 | 5120 |
| gemma-2b | 2.51B | 18 | 2048 |
| gemma-7b | 8.54B | 28 | 3072 |
| stablelm-2-1_6b | 1.64B | 24 | 2048 |
| stablelm-3b-4e1t | 2.80B | 32 | 2560 |
| stablelm-2-12b | 12.14B | 40 | 5120 |
| Mistral-7B-v0.1 | 7.24B | 32 | 4096 |
| mamba-130m-hf | 129M | 24 | 768 |
| mamba-370m-hf | 372M | 48 | 1024 |
| mamba-790m-hf | 793M | 48 | 1536 |
| mamba-1.4b-hf | 1.37B | 48 | 2048 |
| mamba-2.8b-hf | 2.77B | 64 | 2560 |

Table A.2: **Per ROI tests of the interaction between hemisphere and model size**. The first column is the name of each ROI. The second column gives the p-value of the two sample t-test between the set of slopes of all the voxels of the left vs. the right corresponding ROI, whose means and 95% confidence intervals are depicted in Fig. 7b. The third column presents the correlation between the difference in performance of the models (the $r$-scores) on the left hemisphere minus the right hemisphere as a function of the logarithm of the number of parameters, as described in Fig. 7c. The last column gives the p-value of the correlation. These two measures of interaction lead to the same conclusions: the interaction is not significant for BA44 and TP (marginal in the latter case), and highly significant for all the other ROIs, namely aSTS, pSTS, AG_TPJ, BA45 and BA47.

| ROI | p-value of two sample t-test | correlation L-R vs. $\log(n_{\mathrm{parameters}})$ | p-value |
| --- | --- | --- | --- |
| TP | 4.1e-02 | 0.36 | 6.2e-02 |
| aSTS | 7.3e-11 | 0.85 | 1.0e-08 |
| pSTS | 3.5e-09 | 0.82 | 1.2e-07 |
| AG_TPJ | 9.3e-22 | 0.84 | 1.7e-08 |
| BA44 | 4.8e-01 | 0.15 | 4.6e-01 |
| BA45 | 5.7e-07 | 0.73 | 9.4e-06 |
| BA47 | 1.4e-05 | 0.56 | 1.9e-03 |

# B    Supplementary Figures

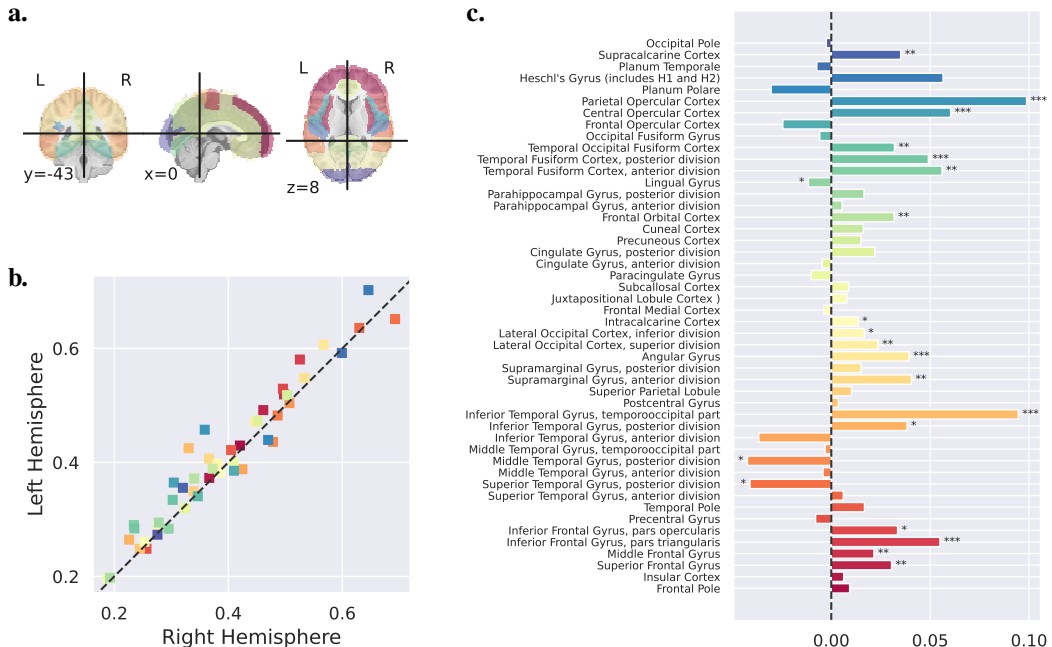

Figure B.1: **Inter-subjects correlations (model free) in homologous cortical regions of the left and right hemispheres**. (a) The 48 cortical areas from nilearn's `cort-maxprob-thr0-2mm` version of the Harvard-Oxford atlas. (b) Relationship between right and left inter-subjects correlations (dots correspond to regions from the atlas) (c) Differences in correlations between left and right regions. The number of stars next to each bar indicates an estimate of the level of statistical significance, assessed with a two-sample t-test (∗:$p < 0.05$, ∗∗:$p < 0.01$, ∗∗∗:$p < 0.001$).

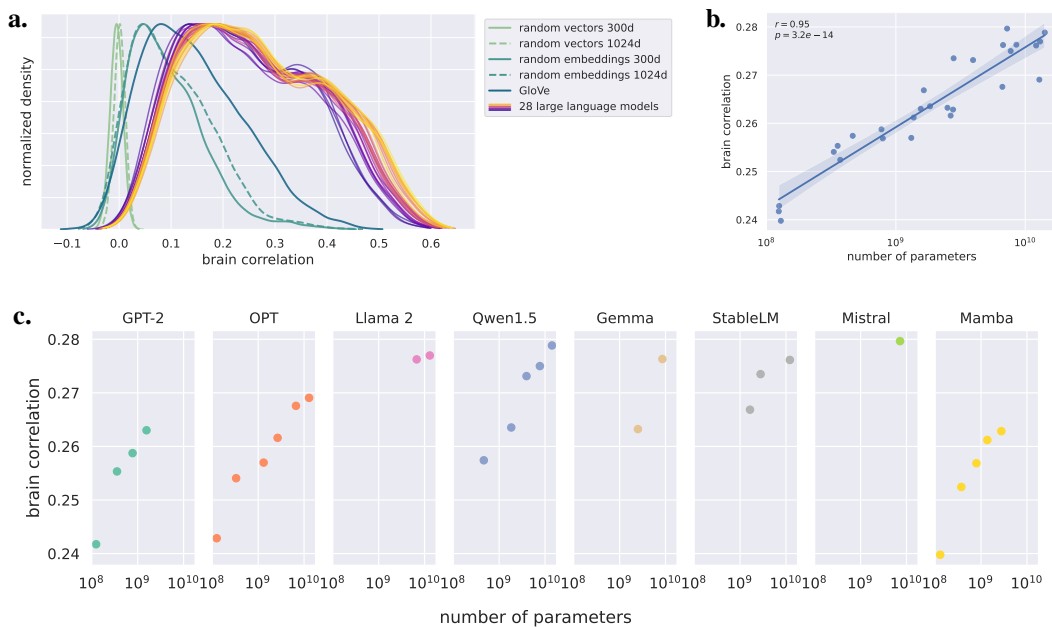

Figure B.2: **Performance of various models in predicting fMRI brain time-courses**. Same as Fig. 2, but for the whole brain volume.

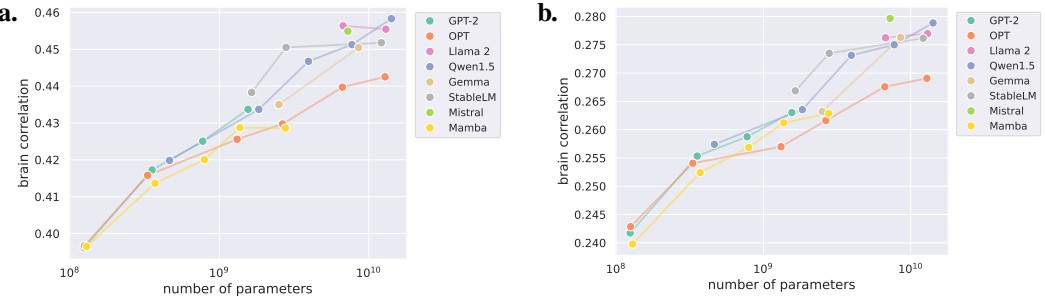

Figure B.3: **Scaling law of fMRI encoding models, by family**. (a) Average over the 25% most reliable voxels (b) Average over the whole brain volume.

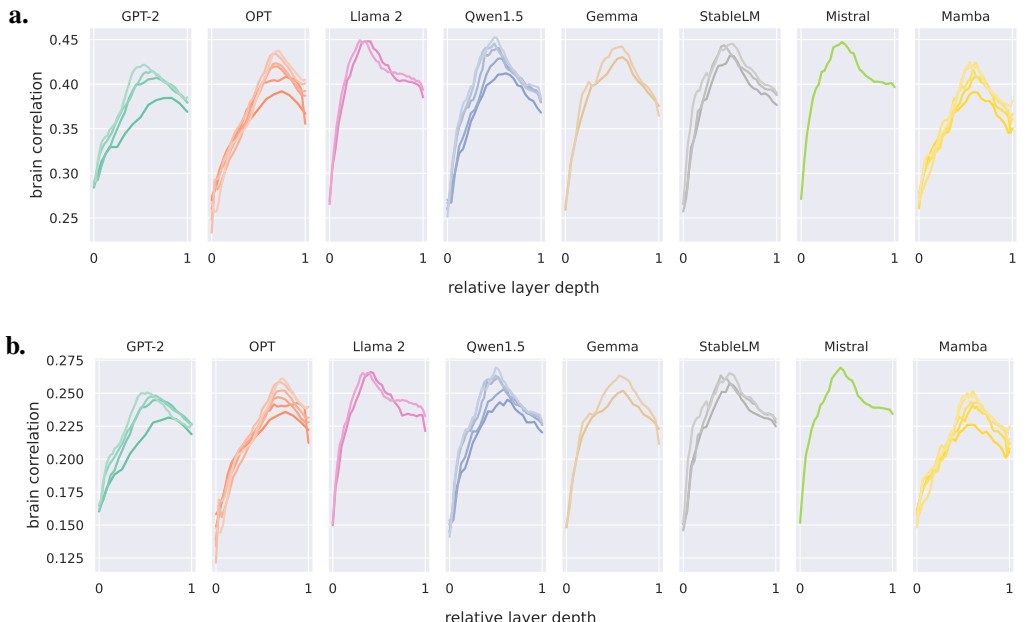

Figure B.4: **Brain correlation as a function of the relative layer depth, split by family**. The relative layer depth corresponds to the depth of a given layer relative to the input and output layers: the first layer is the non-contextual embedding layer, at position 0, and the last hidden layer is at position 1. Considering (a) the 25% most reliable voxels or (b) the whole brain volume.

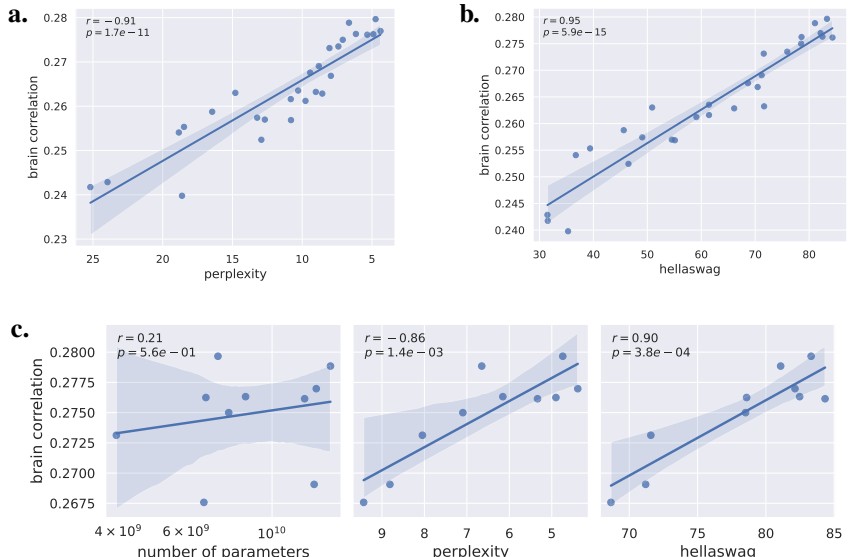

Figure B.5: **Brain correlation and performance on natural language tasks**. Same as Fig. 5, but for the whole brain volume.

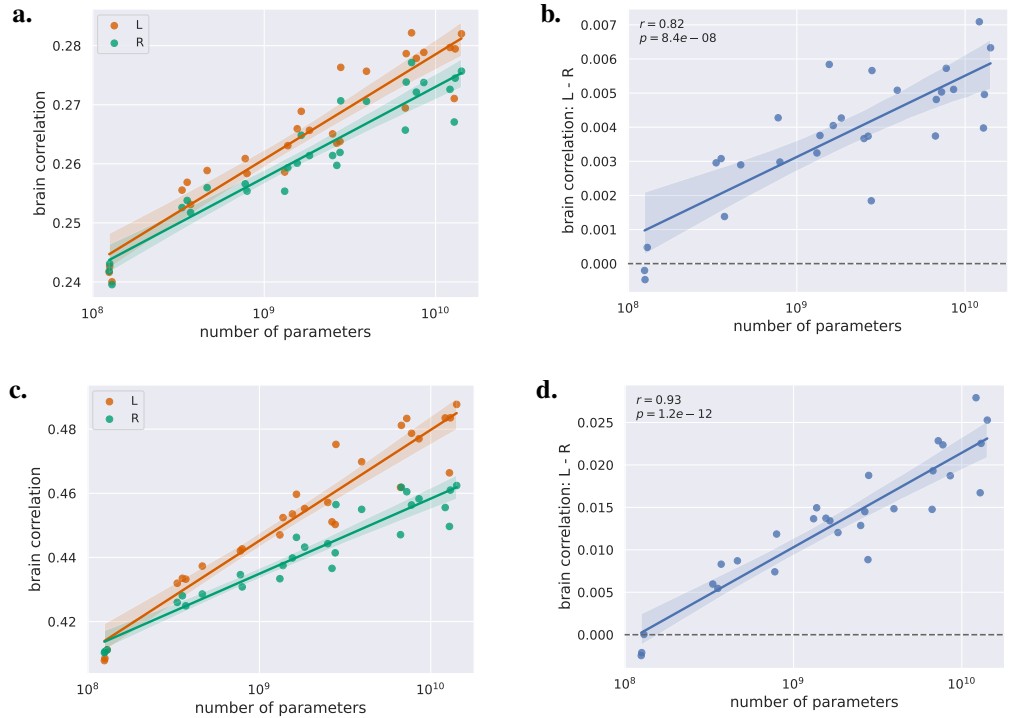

Figure B.6: **Emergence of left lateralization with the size of the encoding models**. Same as Fig. 6, but, top panels (a) and (b), for the whole brain volume, and bottom panels (c) and (d) considering the best 10% voxels of the left hemisphere vs. the best 10% voxels of the right hemisphere.

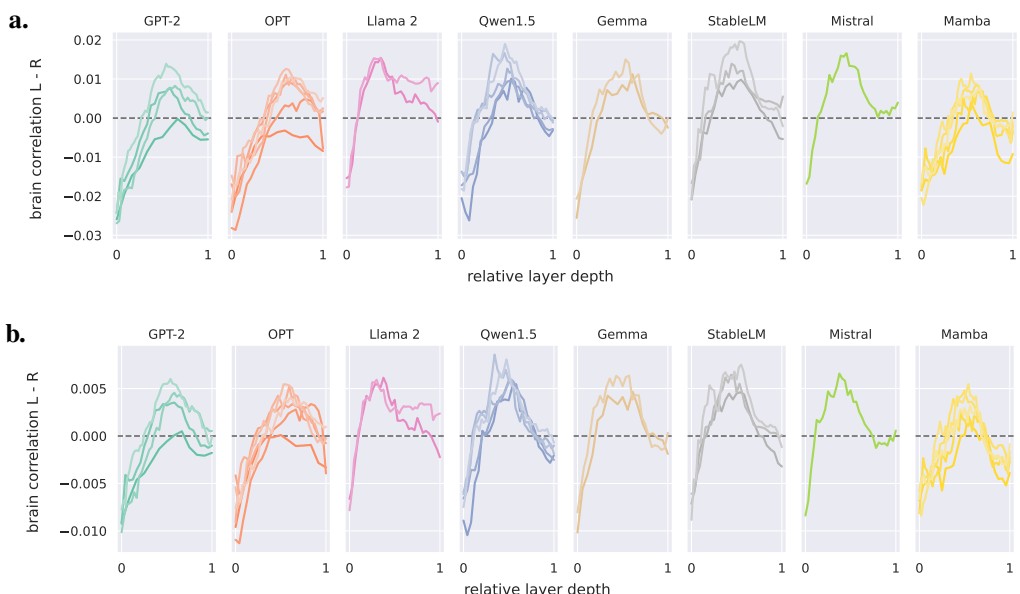

Figure B.7: **Left-right difference in brain correlation as a function of the relative layer depth, split by family**. Considering (a) the 25% most reliable voxels or (b) the whole brain volume.

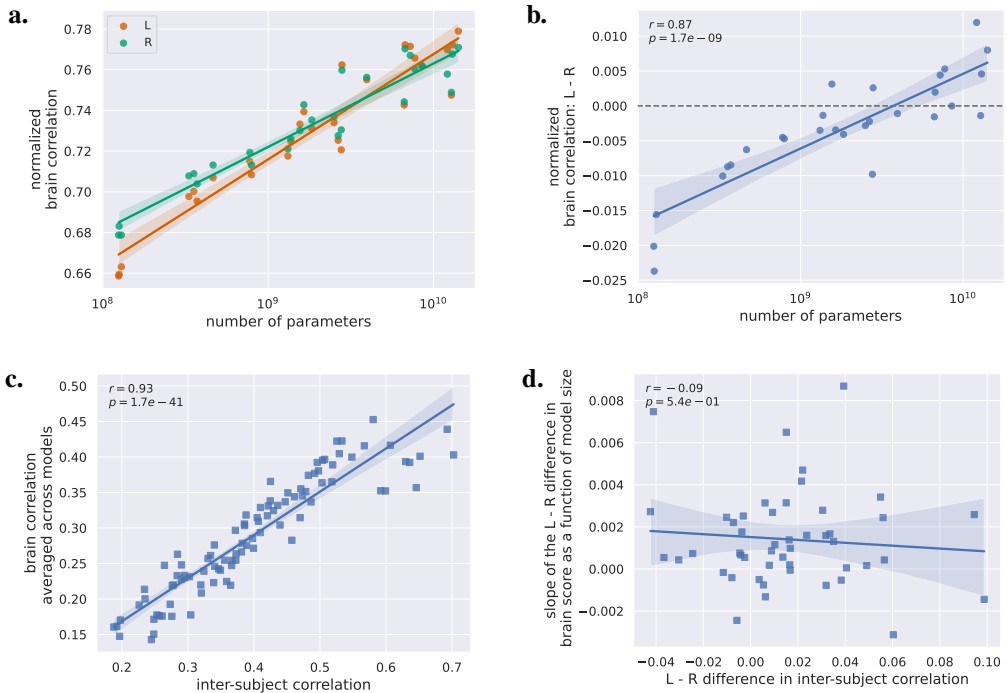

Figure B.8: **Influence of inter-subject correlations on brain scores and growth in asymmetry.** (a) and (b): Same as Fig. 6, but brain correlations are normalized by inter-subject correlations. (c) Brain correlation (average over all models) as a function of the inter-subjects correlation in each of the 48 parcels of the Harvard-Oxford atlas depicted in Fig. B.1. (d) Slope in the relationship between the L-R difference in r-score and the logarithm of the number of parameters, as a function of inter-subjects correlation. Each square corresponds to a parcel in the Harvard-Oxford atlas.

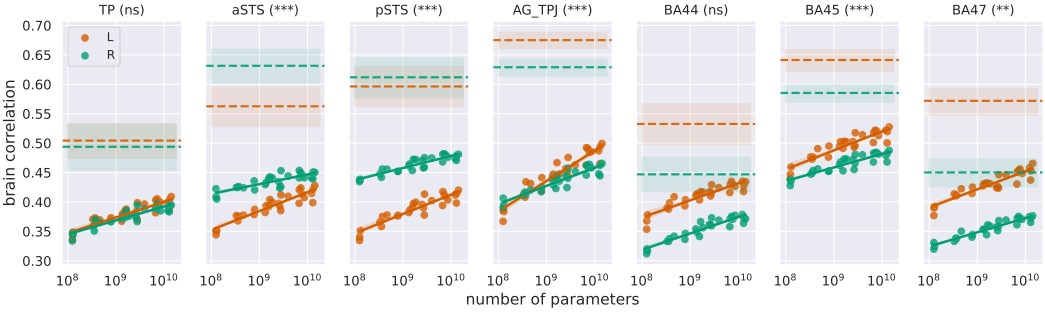

Figure B.9: **Analysis by Regions of Interest.** Brain correlation as a function of the number of parameters (in log scale), for all 28 models, for each ROI, for each hemisphere. The horizontal dashed lines represent the mean inter-subjects correlation in each corresponding ROI (along with 95% confidence interval).

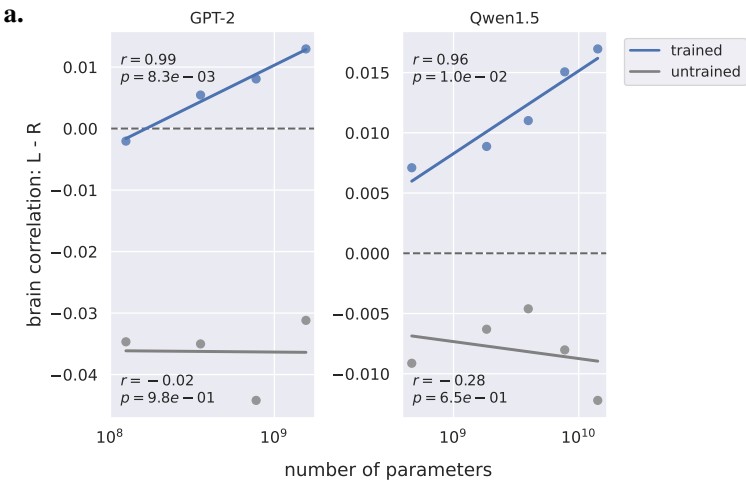

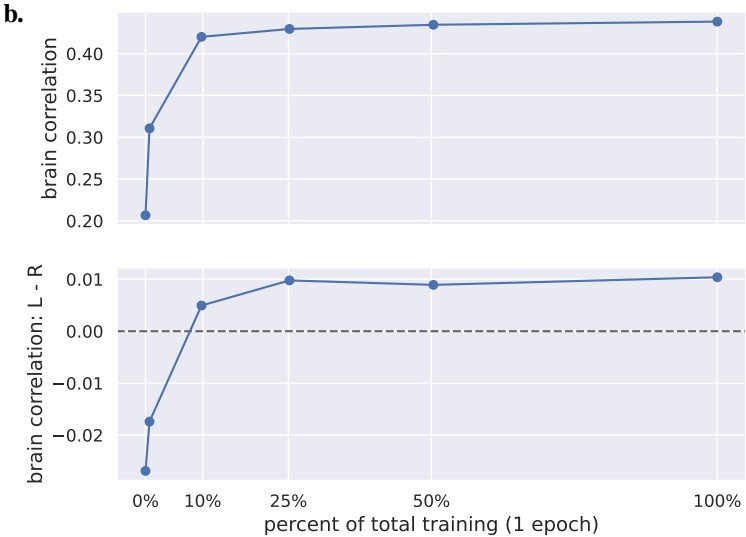

Figure B.10: **Impact of model training**. (a) Left-right difference in brain correlation as a function of the number of parameters (in log scale), for two model families (GPT-2 and Qwen1.5), before (gray dots) and after (blue dots) training. (b) Brain correlation (top) and left-right difference in brain correlation (bottom) as a function of the amount of training, for the Pythia-6.9b model. From the untrained model (0%) to the fully trained model (100% corresponds to one epoch through the 300B tokens training set). All results were computed on the 25% most reliable voxels.

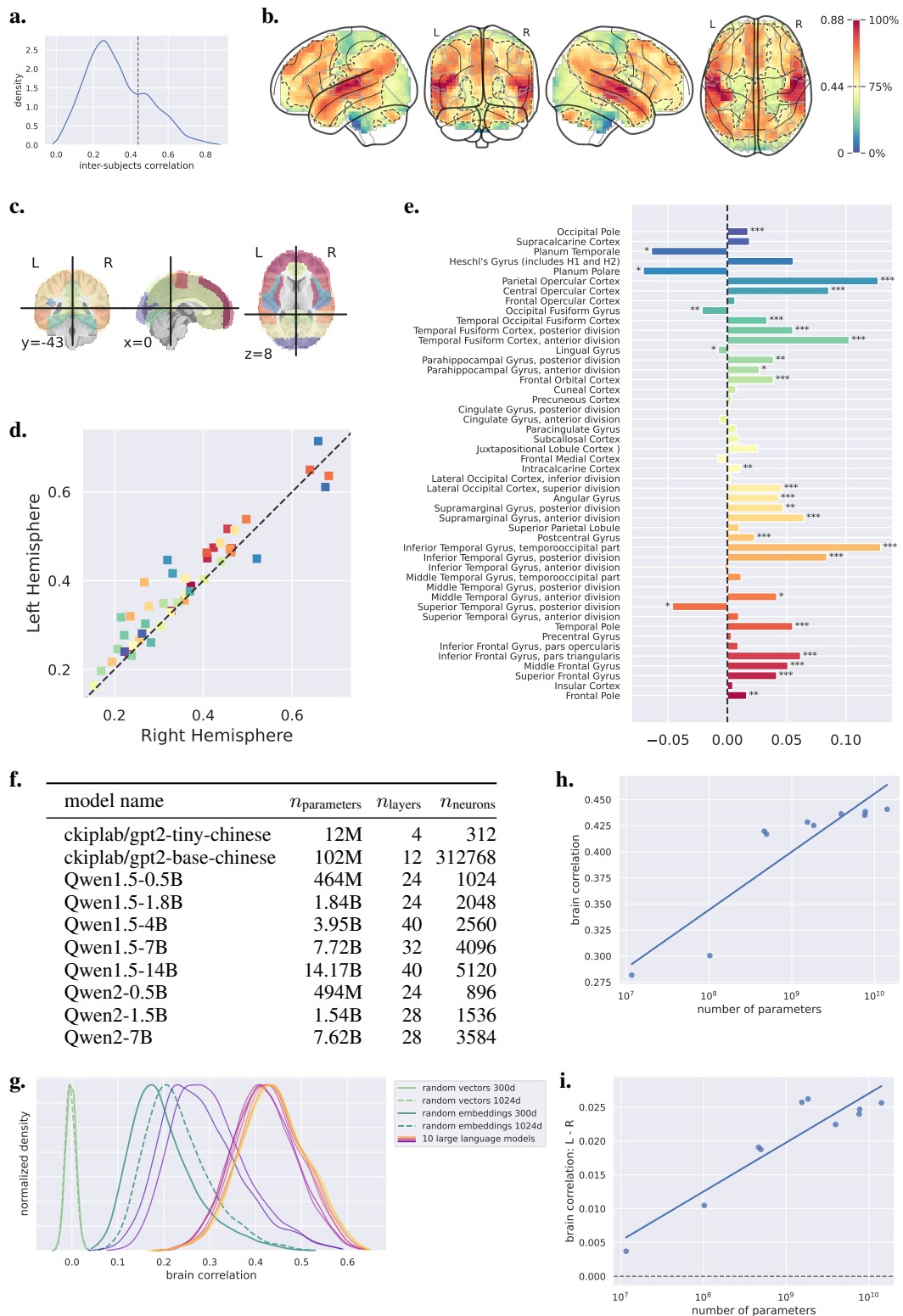

Figure B.11: **Results from Chinese participants and large language models trained on Chinese data**. Analyses performed on all 35 Chinese participants from the multilingual fMRI dataset *Le Petit Prince*. The average subject has 26050 voxels. (a,b) Same as Fig. 1 (a,b). (c,d,e) Same as Fig. B.1 (a,b,c). (f) List of all large language models used in this Chinese study (pretrained with an autoregressive objective on Chinese language data), as available on the Hugging Face hub. (g,h) Same as Fig. 2 (a, b). (i) Same as Fig. 6 (b).

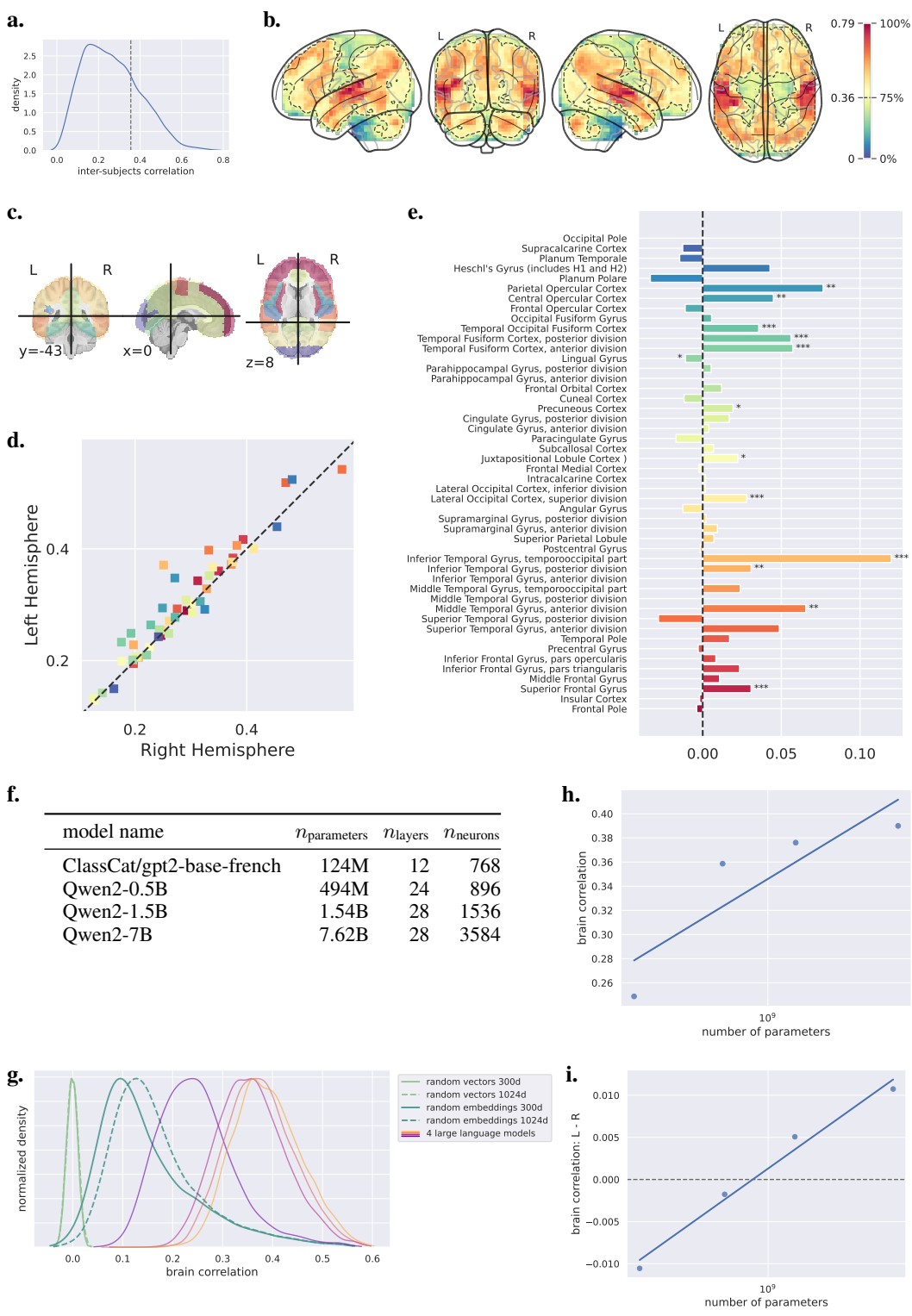

Figure B.12: **Results from French participants and large language models trained on French data**. Analyses performed on all 28 French participants from the multilingual fMRI dataset *Le Petit Prince*. The average subject has 27773 voxels. (a,b) Same as Fig. 1 (a,b). (c,d,e) Same as Fig. B.1 (a,b,c). (f) List of all large language models used in this French study (pretrained with an autoregressive objective on French language data), as available on the Hugging Face hub. (g,h) Same as Fig. 2 (a, b). (i) Same as Fig. 6 (b).

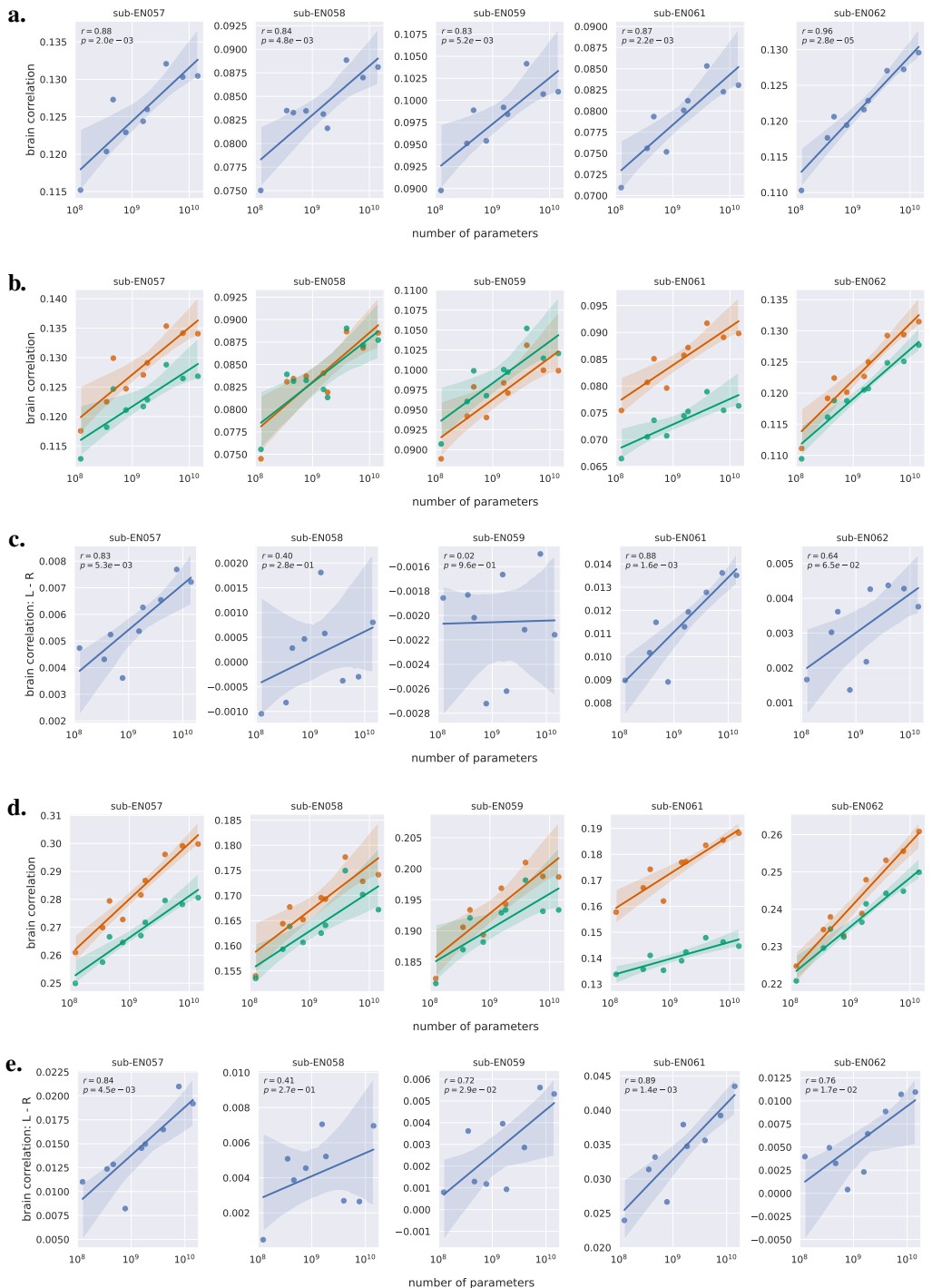

Figure B.13: **Individual analyses performed on five English participants**. Each column corresponds to one participant, `sub-EN057`, `sub-EN058`, `sub-EN059`, `sub-EN061` and `sub-EN062` respectively. Nine large language models were used: all the models from the GPT-2 and Qwen1.5 families used in the main study (see Table A.1), ranging from 124M to 14.2B parameters. (a) For each individual, brain correlation on the whole brain volume as a function of the number of parameters (in log scale). (b) Same as (a), but for the left hemisphere (red) and for the right hemisphere (green). (c) Difference between left and right hemispheric brain correlations. (d) and (e) Same as (b) and (c) but considering the best 10% voxels of the left hemisphere vs. the best 10% voxels of the right hemisphere.

