# OpenReview forum: "fMRI predictors based on language models of increasing complexity recover brain left lateralization"
_NeurIPS.cc/2024/Conference — NeurIPS 2024 poster_

### Official Review · Reviewer_LDde · 2024-06-22

**Soundness:** 2
**Presentation:** 3
**Contribution:** 2
**Rating:** 6
**Confidence:** 4

**Summary:**

The paper studies how well 28 large language models (LLMs) predict fMRI activity of human subjects listening to an audiobook. First, they observe a scaling law; the neural predictivity of LLMs increases linearly with the logarithm of the number of model parameters, a result consistent with prior work. Second, they show larger models predict left-hemisphere activity better than right-hemisphere activity, and this left-right asymmetry increases with model size.

**Strengths:**

1. The paper is written clearly and has put effort into sufficiently explaining the methodological details.

**Weaknesses:**

From most to least significant:
1. For the paper's main contribution, i.e., asymmetry in Left-Right neural predictivity that increases with model size (Figure 6), the paper does not provide sufficient experimental/theoretical analyses into possible reasons, nor an adequate attempt into possible explanations/interpretations.
2. The growing L-R asymmetry result may suggest qualitatively interesting differences between larger vs smaller LLMs, as briefly mentioned in Lines 256-260 in the paper. However, an alternative reason could be that right-hemisphere activity is just lower in general (as shown in many prior studies), or more noisy / less consistent across subjects (especially since this paper uses a group-average of fMRI activity).
- The L-R difference does not seem qualitatively interesting, because both trends in Fig. 6a are straight lines. This seems to simply hint at left-hemisphere activity being easier to predict. If the left-hemisphere line were exponentially increasing instead, that may hint at a qualitatively interesting difference between larger vs smaller models. But instead, we see two straight lines. In fact, the L-R difference seems to already be present with models with 350M parameters, and the gap is only linearly widening as model size increases.
3. For the growing L-R asymmetry result, there is no significant effect in many key regions of the language network (IFG, temporal pole) (Figures 4 and 7) (Lines 225-227). The strongest effects seem to occur in AG, which some consider not part of the language network [1]. This is a relevant weakness since this paper focuses on the language network.
4. They compute a group average of fMRI activity across subjects before using LLMs to predict fMRI activity (Line 64), and identified regions of interest (ROIs) in the language network using anatomical locations (Line 95-98), rather than functional localization. However, these methodological choices may have issues when studying the language network (see Box 1 of [1]). Because of inter-individual variability in the precise anatomical locations and sizes/shapes of functional areas in the language network, any voxel defined in a common anatomical space often corresponds to different functional areas across individuals. Hence, using group averages and anatomical ROI localization can lead to the blurring of neighboring areas and information loss [1].

[1] E. Fedorenko, “The language network as a natural kind within the broader landscape of the human brain,” Nature Reviews Neuroscience.

**Questions:**

Just a minor suggestion to fix typos:
1. Hyperlinks to appendix figures do not work correctly. E.g., "Fig. B.1" (Line 162), also: Lines 173, etc. These are the hyperlinks that start with "Fig B."

**Limitations:**

Limitations not mentioned in the paper:
1. See Weaknesses 3 and 4.

---

> ### Author Rebuttal · Authors · 2024-08-05
>
> Thank you for your feedback. Here is a point-by-point response to your comments.
>
> > The growing L-R asymmetry result may suggest qualitatively interesting differences between larger vs smaller LLMs, as briefly mentioned in Lines 256-260 in the paper. However, an alternative reason could be that right-hemisphere activity is just lower in general (as shown in many prior studies), or more noisy / less consistent across subjects (especially since this paper uses a group-average of fMRI activity).
>
> Thanks for raising the point of left vs right signal-to-noise in the data. Indeed, one intuitively expects higher brain correlations in regions where the signal-to-noise ratio is better and therefore predicts left-right asymmetries based on this point. First of all, note that here, we compare a large variety of models using the same data, so that the quality of the data is in fact a constant across models. This is different from the previous study (Caucheteux & King, 2022) that reported a left-right asymmetry using a single model. To investigate this issue further, we performed new analyses that we intend to include as supplementary figures. They demonstrate that while the average r-scores (across model sizes), is related to the inter-subjects correlations (ISC), this is not the case for the slope in the relationship between brain correlation and model size.
>
> Plots from Fig.1 in the rebuttal pdf display the left and right, model-free, inter-subjects correlations (ISC)  in various brain regions from the Harvard-Oxford atlas. Some regions, especially in the language network, indeed show asymmetries. Fig. 2a, based on Fig. B5 of the current paper, shows ROI analyses with the addition of horizontal lines indicating the region’s ISCs. Fig. 2b displays the relationship between the average r-score and the ISC, and Fig. 2c the relationship between the slope of r-scores as a function of the number of parameters, and the ISC. Consider for instance aSTS and BA44: in the aSTS, the inter-subjects reliability is greater in the right hemisphere, and on average so is the r-score of the encoding models, but the slope in the relationship between brain correlation and the size of the models is greater in the left aSTS; conversely, in BA44, although the inter-subjects correlation is higher in the left hemisphere, which is reflected in location of the brain correlation of the models on the y-axis, the slopes are quite similar.
>
> > The L-R difference does not seem qualitatively interesting, because both trends in Fig. 6a are straight lines. [...] If the left-hemisphere line were exponentially increasing instead, that may hint at a qualitatively interesting difference between larger vs smaller models. But instead, we see two straight lines. In fact, the L-R difference seems to already be present with models with 350M parameters, and the gap is only linearly widening as model size increases.
>
> It is not clear to us why only nonlinear effects would be interesting. Perhaps the idea is that steps in the relationship between number of parameters and brain scores would reveal the  sudden emergence of new capacities for the network (?). This is not what we observed, and in that sense, it means that we learn something from the data.
>
> > For the growing L-R asymmetry result, there is no significant effect in many key regions of the language network (IFG, temporal pole) (Figures 4 and 7) (Lines 225-227). The strongest effects seem to occur in AG, which some consider not part of the language network [1]. This is a relevant weakness since this paper focuses on the language network.
>
> 5 out of the 7 ROIs actually show significant effects (and 2 of them are located in the IFG). Regarding the Angular Gyrus (AG), while it does not often show up in language minus control contrasts and is thus not included in the core language network by some researchers, it has long been known to be sensitive to linguistic operations, for example, semantic composition (Price et al. 2015 cited in our paper). This is the rationale why we included it in the a priori list of ROIs (before looking at the data!). We are not making a strong statement about what regions belong or not to the language network here, but we think that the inclusion of the Angular Gyrus is reasonable.
>
> The fact that the slopes do not differ in a BA44 is a weakness only if one a priori considers that it should show an asymmetry. The role of BA44 is actually fiercely debated and some authors consider it to be involved in bilateral articulatory/phonological processes (Matchin & Hickok, 2020, Cereb. Cortex). So rather than a weakness, we take the BA44 result, if it is replicated, as an empirical fact to be explained.
>
> > Because of inter-individual variability in the precise anatomical locations and sizes/shapes of functional areas in the language network, any voxel defined in a common anatomical space often corresponds to different functional areas across individuals. Hence, using group averages and anatomical ROI localization can lead to the blurring of neighboring areas and information loss
>
> This remark is correct. The Little Prince fMRI dataset does not contain a language localizer that would have permitted us to identify the most sensitive voxels at the individual level. However, given the size of our ROIs and the smoothness of the data we are working with, our experience is that selecting individual voxels from a functional localizer has very little impact. Without doubt, ROIs analyses, by averaging over voxels with different functional profiles, blur the results. Actually, any kind of averaging between subjects has this issue. In our opinion, the ROIs analyses are still a worthy complement to a voxel-based view, that we present on Fig.4.
>
> > Hyperlinks to appendix figures do not work correctly. E.g., "Fig. B.1" (Line 162), also: Lines 173, etc. These are the hyperlinks that start with "Fig B."
>
> Thanks for pointing this out, it is corrected in the revised version.

---

> > ### Comment · Reviewer_LDde · 2024-08-09
> >
> > I raised my score from 4 to 5. Thanks for the clarifications and new rebuttal pdf results. The authors provided convincing clarifications for some of the weaknesses (and their sub-points) I raised.
> >
> > 1. The paper does not provide sufficient experimental/theoretical analyses into possible reasons for growing L-H asymmetry, nor an adequate attempt into possible explanations/interpretations (unresolved)
> > 2. Linearly growing L-R asymmetry may simply reflect L-R differences in SNR (partially resolved)
> > - In my review, I wrote that straight lines in Fig. 6a (x-axis: model size, y-axis: brain correlation) may not be interesting, even though the LH slope is steeper than the RH slope. In particular, I was thinking that this may simply reflect LH having a higher SNR / noise ceiling than RH. In this case, if you changed the y-axis to the noise-normalized brain correlation (i.e., brain correlation divided by respective noise ceilings), the LH and RH slopes would be equal. Here, the growing asymmetry simply reflects that LH activity is easier to predict, rather than an interesting difference between larger vs smaller models.
> > - Put another way, if you had two datasets where one had a higher noise ceiling, a "growing asymmetry" or difference in model-size-vs-brain-correlation slopes between the two datasets is not interesting if it is simply due to the difference in noise ceilings.
> > - Even after looking at the new rebuttal pdf results, I am unsure if the L-H difference in SNR fully or only partially explains the L-H asymmetry in slopes.
> > 3. Some regions in the language network do not show a significant effect (resolved)
> > - Thanks for the clarification.
> > 4. Anatomical rather than functional localization, and group-level analysis (mostly resolved)
> > - Thanks for the clarification.

---

> > > ### Author Response · Authors · 2024-08-12
> > >
> > > Thank you for your constructive feedback.
> > >
> > > > In this case, if you changed the y-axis to the noise-normalized brain correlation (i.e., brain correlation divided by respective noise ceilings), the LH and RH slopes would be equal.
> > >
> > > Maybe we misunderstand something here. If one considers the inter-subjects correlation (ISC) as a proxy for noise ceiling, then it is a constant from the data that does not depend on the model, in particular in the number of parameters. In that case, dividing the brain correlations by the ISC would affect the global position on the y-axis but not the ratio between the slopes.
> > >
> > > > I am unsure if the L-H difference in SNR fully or only partially explains the L-H asymmetry in slopes.
> > >
> > > In our response to reviewer d74H, we report a new analysis that goes against the idea that the growing left-right asymmetry in brain correlation is explained by an asymmetry in inter-subjects correlation. We hope that it answers your concern.

---

> > > > ### Comment · Reviewer_LDde · 2024-08-13
> > > >
> > > > I will raise my score from 5 to 6. I think my query was mostly resolved by your new clarification, combined with your clarification to reviewer d74H and Fig. 2c in your rebuttal pdf. Below, I will try my best to clarify what I meant in my earlier comment.
> > > >
> > > > > dividing the brain correlations by the ISC (noise ceiling) would affect the global position on the y-axis but not the ratio between the slopes.
> > > >
> > > > Maybe I am the one who is misunderstanding, but I think the ratio between the LH/RH slope values is affected by the LH/RH noise ceilings.
> > > >
> > > > A toy example, suppose:
> > > > - LH values: [2,4,6,8,10], if the noise ceiling is 10, after normalization division: [0.2, 0.4, 0.6, 0.8, 1.0].
> > > > - RH: [1,2,3,4,5], if the noise ceiling is 5, after normalization division: [0.2, 0.4, 0.6, 0.8, 1.0].
> > > > - The ratio between the steepness of slopes started as 2, but changed to 1, after dividing by the different noise ceilings.
> > > >
> > > > Specifically, if the ratio of the un-normalized LH/RH slopes (current Fig. 6a) is equal to the ratio of the LH/RH noise ceilings, the normalized LH/RH slope values (i.e., divided by noise ceilings) will end up being equal.
> > > > - Slope ranges for LH: [0.402, 0.465], RH: [0.398, 0.448], I eyeballed the values from Fig. 6a, exact values are not important.
> > > > - Thus, the ratio of slope: (0.063 / 0.050)
> > > > - Suppose the noise ceiling values are: LH: 0.700, RH: 0.556, which has a ratio: (0.700 / 0.555...), ~equal to (0.063 / 0.050).
> > > > - Then, the noise-normalized slope ranges for LH: [0.574, 0.664], RH: [0.716, 0.806]
> > > > - Thus, the ratio of LH/RH slopes: (0.09 / 0.09) = 1, which means the LH/RH slope values become equal after ceiling normalization
> > > >
> > > > Perhaps it is best to show a figure (maybe in the appendix) similar to Fig. 6a in your paper, but the y-axis is the noise-normalized brain correlation instead. The LH slope is probably still steeper than the RH slope, as you said, and this would resolve my query.

---

> > > > > ### Author Response · Authors · 2024-08-13
> > > > >
> > > > > Thank you for the example. We now understand better and you are perfectly right that the normalization by the ISC can modify the ratio of slopes. We therefore recomputed Fig.6 (a and b) by replacing the brain scores by the brain scores divided by the ISC, in each voxel (and then averaged by hemisphere). The growing asymmetry still holds with these normalized values, and the left-right difference still follows a scaling law very similar to the one displayed in Fig. 6b with, now, r=0.87, p=1.7e-9 (while before normalization, r=0.89; p=1.8e-10). We will include this figure in the future appendix as suggested. Thanks again for helping us to strengthen the paper!

---

### Official Review · Reviewer_d74H · 2024-07-07

**Soundness:** 3
**Presentation:** 4
**Contribution:** 3
**Rating:** 7
**Confidence:** 5

**Summary:**

The authors investigate whether larger-parameter language models better predict left versus right hemisphere brain responses (recorded during listening of a naturalistic story, via fMRI), motivated by left-lateralized processing of language in most individuals. They indeed find that larger models better predict left hemisphere brain responses, compared to right hemisphere responses. In addition, the authors have two separate findings that mainly replicate prior work: larger models better predict brain responses, and the prediction performance correlates with the language model's ability to perform natural language tasks.

**Strengths:**

- The primary finding of the paper, that larger language models better predict left hemisphere brain responses, is novel.
- The paper is well-written and well-cited. The structure of the paper is intuitive, making it easy to follow.
- The paper has a good number of control experiments.
- The paper provides nice replication of former work in this domain.

**Weaknesses:**

- The paper averages across different individual's brain responses (n=48) in the template brain space. Individual activations to language differ from participant to participant, and there is no direct voxel-to-voxel correspondence (see e.g., https://pubmed.ncbi.nlm.nih.gov/32160565/). Hence, it is somewhat unclear whether the primary finding of brain-model lateralization can be explained away by methodological issues. Imagine that individuals generally have a higher voxel-to-voxel correspondence in the left hemisphere compared to the right -- would it be true that large models, given higher expressivity, would be able to better predict those left-hemisphere voxels? (see Questions for more specific questions).

**Questions:**

- Pertaining to the issue of group-level averaging, is it true that the left-hemisphere generally has higher reliability? Was that controlled for in any way in the analyses (as far as I understand, the authors analyze the top 25% most reliable voxels, yielding approximately a similar number of voxels in both hemispheres (~3K). What are the average reliability values per hemisphere? If the authors subset left and right hemisphere voxels such that they are matched on reliability, are the primary model-brain lateralization findings still valid?
- Are the performance vs. brain correlations (Figure 5) equally strong in both hemispheres? Do the left hemisphere brain activations contain more information about task, relative to the right ones?
- Can the authors clarify what they mean by "The right hemisphere is usually “hidden” because it is inhibited by the left hemisphere in healthy people"? (line 285)?

**Limitations:**

The authors do discuss limitations briefly, but I am not sure I agree with the conclusions that follow. For instance, in lines 299-300 the authors mention the group-level approach, but conclude that assessing inter-individual variability would require "However, this would require a random sample of the population, not only right-handed participants". Why is that true? Inter-individual variability can still be investigated in right-handed individuals? It is true that it would be useful to assess the hemispheric dominance of individuals -- that could be done by e.g., assessing the consistency of brain responses within an individual, across runs (other approaches exist as well).

---

> ### Author Rebuttal · Authors · 2024-08-05
>
> Thank you for your feedback. Here is a point-by-point response to your comments.
>
> > Pertaining to the issue of group-level averaging, is it true that the left-hemisphere generally has higher reliability? Was that controlled for in any way in the analyses (as far as I understand, the authors analyze the top 25% most reliable voxels, yielding approximately a similar number of voxels in both hemispheres (~3K). What are the average reliability values per hemisphere? If the authors subset left and right hemisphere voxels such that they are matched on reliability, are the primary model-brain lateralization findings still valid?
>
> Thanks for bringing this up. The description of the data itself deserves more attention, and the question of whether the effect is simply due to a difference in the left-right signal-to-noise ratio is an important one. We will discuss these two aspects in more detail in the camera-ready version, along with the addition of two new figures shared in the rebuttal pdf that show new analyses.
>
> First, with respect to the question of group averaging, we show that the left hemisphere does indeed have a higher reliability in general (see Fig. 1 of accompanying pdf), although the inter-subjects reliability follows the diagonal in left vs. right plot as a general trend. Now, importantly for the discussion, this could explain why for a single encoding model there is a better correlation in the left than in the right hemisphere, but this could not explain the left-right difference scaling law as a function of the number of parameters, since the quality of the data is a constant across models. In short, Fig. 2 shows that (i) the mean brain correlation follows the reliability of the signal in the different regions of interest, but that (ii) the slopes do not generally follow such a trend. This point is also discussed at length in another reviewer thread.
>
> > Are the performance vs. brain correlations (Figure 5) equally strong in both hemispheres? Do the left hemisphere brain activations contain more information about task, relative to the right ones?
>
> The relationships between performance (hellasawag or perplexity) and r-scores, split by hemisphere, show the same increasing left-right difference as the version with (log of) number of parameters on the x-axis, that is, the better the performance, the more marked the asymmetry. These graphics will be added as supplementary figures of the paper.
>
> More generally, we think that it is important to further investigate the relationship between the performance on a given task (to name but a few: syntactic comprehension, sentiment analysis, social understanding) and the difference in brain score in a given hemisphere or region. We are working on this, but this goes well beyond the present paper.
>
> > Can the authors clarify what they mean by "The right hemisphere is usually “hidden” because it is inhibited by the left hemisphere in healthy people"? (line 285)?
>
> This affirmation refers to the concept of interhemispheric inhibition according to which one hemisphere can prevent concurrent processing by the opposite hemisphere. It is one of the theories proposed to explain the capacity of the right hemisphere to take over linguistic functions in aphasic patients with left lesions (for example, Tzourio-Mazoyer et al., 2017, cited in our paper, wrote:  “In typical brains, inter-hemispheric inhibition, exerted from the LH to the RH, permits the LH to maintain language dominance. In pathological conditions, inter and intra-hemispheric inhibition is decreased, inducing modifications on the degree of Hemispheric Specialisation and of language networks.”). We will tone down this assertion to make it clearer that it is an hypothesis.
>
> > The authors do discuss limitations briefly, but I am not sure I agree with the conclusions that follow. For instance, in lines 299-300 the authors mention the group-level approach, but conclude that assessing inter-individual variability would require "However, this would require a random sample of the population, not only right-handed participants". Why is that true? Inter-individual variability can still be investigated in right-handed individuals? It is true that it would be useful to assess the hemispheric dominance of individuals -- that could be done by e.g., assessing the consistency of brain responses within an individual, across runs (other approaches exist as well).
>
> It is true that interindividual variability could be investigated only in right-handed participants. What we had in mind is that if one wanted to assess if the LLMs correlate more strongly with the dominant hemisphere of humans in general, it  would make sense to generalize to the whole population, rather than a sub-population. Anyway, because this issue is not central to the paper, we will remove the sentence. A more relevant focus for this section is “what could we learn from individual analyses?”, and we will expand on that.

---

> > ### Comment · Reviewer_d74H · 2024-08-11
> >
> > Thanks to the authors for addressing my comments, in particular, for running the analyses in Figure 2. I think the claims need to be rephrased slightly by taking those analyses into account. I still struggling to make sense of Figure 2B. Usually, larger models also perform better on the tasks (which also correlates with brain scores, as you demonstrate). Then why would it not be the case that the number of parameter trend holds up? Any speculations are welcome. Thanks again!

---

> > > ### Author Response · Authors · 2024-08-12
> > >
> > > Thank you for your constructive feedback. We will include and discuss these new analyses.
> > >
> > > >  Then why would it not be the case that the number of parameter trend holds up? Any speculations are welcome.
> > >
> > > We understand the concern. The relationship between inter-subjects correlation (ISC) and performance of encoding models is indeed complex and subtle.
> > >
> > > We produced a new plot that might answer your concern (which we unfortunately cannot share here, but will include in supplementary figures). This plot shows, for each (pairwise symmetric) parcel of the Harvard-Oxford atlas, the left-right difference in inter-subject correlation on the x-axis, and, on the y-axis, the slope of the left-right difference in brain score as function of model size (this corresponds to the slope of the line displayed in Fig. 6b, computed over the whole brain in this figure, but computed now for each parcel) . The verdict is that there is no relationship between the two quantities (r = -0.1; p=0.52).

---

> > > > ### Comment · Reviewer_d74H · 2024-08-13
> > > >
> > > > Thanks for this additional piece of information. I am still puzzled as to why that is the case (perhaps due to an aspect of the evaluation pipeline?). However, I think the paper has improved substantially over rebuttals, and I am happy to raise my score to 7.

---

### Official Review · Reviewer_jQCd · 2024-07-10

**Soundness:** 3
**Presentation:** 3
**Contribution:** 3
**Rating:** 7
**Confidence:** 5

**Summary:**

Recent studies on language processing in the human brain using fMRI data and language model embeddings have shown that both hemispheres are involved in language processing although many previous studies have indicated a left lateralization. The authors aim to reconcile these findings. They use embeddings from language models of different sizes to predict naturalistic fMRI data and show that as the number of parameters or performance of the models increases, the respective embeddings are more predictive of activity in the left hemisphere vs. in the right one. They also show that this pattern holds in various regions of interest.

**Strengths:**

- Originality: The main novel contribution of this work is showing the clear dependence between model capacity or performance and predicted left lateralization of language processing. Many previous studies, including some that have been cited, demonstrate left lateralization to a certain extent (ex: Fig 3 from Cauchetaux and King (2022)) and have shown that predictivity scales with language model capacity (ex: Fig 2 from Schrimpf et al. (2021)). However, I appreciate the thoroughness and clarity of the authors’ analyses and believe that it provides additional evidence to back a key finding in neuroscience.

- Quality: Experiments performed to support claims are very thorough and well-motivated.

- Clarity: The paper is well-written and easy to understand. One should be able to reproduce the results presented using the paper and associated code.

- Significance: As mentioned above, although previous studies have shown left lateralization to a certain extent and that predictivity scales with language model capacity, I believe that this work is still important since it clearly shows that more left-lateralization emerges as the models are scaled up.

**Weaknesses:**

This is a strong submission in general. I have the following suggestions/questions to improve it:

1. The second paragraph of the introduction (and the abstract) mentions that many recent works have discovered strikingly symmetric brain maps for language processing. To the best of my knowledge, many of these studies conclude that both hemispheres are involved in language processing without assessing left-lateralization since other studies had contended the role of the right hemisphere altogether. If one looks at their figures (ex: Fig 2 from Toneva and Wehbe (2019) or Fig 3 from Cauchetaux and King (2022)), the left hemisphere is generally predicted better than the right hemisphere. Therefore, I think that this paragraph can be modified to better reflect these results.

2. I understand the need for reducing the computational burden of the study and the nuances mentioned in the limitations section. However, for a subset of the participants, it would be great to see individual participant-level results to make an even more convincing argument. This is mostly to rule out any artifacts associated with using an average subject and not to make inferences about inter-individual variability.

3. For encoding any given word, are the embeddings computed by only inputting the previous words or using the full sentence/text? In other words, are the embeddings capturing future information due to the full sentence/text being encoded at once? If they are capturing future information, that is at odds with how the data was collected since words were sequentially presented to participants.

**Questions:**

I have mentioned my questions and suggestions in the Weaknesses section above.

**Limitations:**

The authors have identified the limitations of their analyses and I do not foresee any negative societal impacts.

---

> ### Author Rebuttal · Authors · 2024-08-05
>
> Thank you for your feedback. Here is a point-by-point response to your comments.
>
> > The second paragraph of the introduction (and the abstract) mentions that many recent works have discovered strikingly symmetric brain maps for language processing. To the best of my knowledge, many of these studies conclude that both hemispheres are involved in language processing without assessing left-lateralization since other studies had contended the role of the right hemisphere altogether. If one looks at their figures (ex: Fig 2 from Toneva and Wehbe (2019) or Fig 3 from Cauchetaux and King (2022)), the left hemisphere is generally predicted better than the right hemisphere. Therefore, I think that this paragraph can be modified to better reflect these results.
>
> Thanks for pointing that out. We apologize for missing these passages and will modify our introduction accordingly.
>
> > I understand the need for reducing the computational burden of the study and the nuances mentioned in the limitations section. However, for a subset of the participants, it would be great to see individual participant-level results to make an even more convincing argument. This is mostly to rule out any artifacts associated with using an average subject and not to make inferences about inter-individual variability.
>
> We agree this is an important point, but we will not be able to conduct this work in the time-frame. We have modified our paragraph in the Limitations section to better acknowledge the interest of performing the same analyses at the individual level, which we will do in the future.
>
> > For encoding any given word, are the embeddings computed by only inputting the previous words or using the full sentence/text? In other words, are the embeddings capturing future information due to the full sentence/text being encoded at once? If they are capturing future information, that is at odds with how the data was collected since words were sequentially presented to participants.
>
> Indeed, the embeddings are computed autoregressively, taking into account only the past of a given word, and not future information.

---

> > ### Comment · Reviewer_jQCd · 2024-08-09
> >
> > Thank you for the response! My concerns have been addressed but it would be great if the authors could include the individual level analyses in the final version. I keep my original score.

---

> > > ### Author Response · Authors · 2024-08-12
> > >
> > > Thank you for your constructive feedback.
> > >
> > > > it would be great if the authors could include the individual level analyses in the final version
> > >
> > > The full pipeline takes about five days. Before the deadline, we will try and pick a few subjects and maybe a subset of models to check the validity of the results at the individual level.

---

### Official Review · Reviewer_Wj1c · 2024-07-12

**Soundness:** 3
**Presentation:** 1
**Contribution:** 1
**Rating:** 6
**Confidence:** 4

**Summary:**

The paper studies whether encoding fits with LLMs onto fMRI data can be used to find left lateralization, the idea that language is lateralized to the left hemisphere. This is a well-known property of language localization in many humans. They also show that increasing model size improves fit from their encoding models.

**Strengths:**

* A new fMRI dataset that can serve as additional analysis for LLMs.
* Incorporation of a large family of LLMs never studied before in previous work including LLaMA-2, Mamba, etc.
* Likely the first (to my knowledge) to directly test the correspondence between LLMs and lateralization.

**Weaknesses:**

* Although the motivation of this paper is reasonable, I think the presentation can be significantly improved.
    * The paper should make an argument about why lateralization is an important property of brain fits with LLMs. I think this was missing.
    * A small nitpick: From my understanding, I can’t see any data from right hemisphere regions in Schrimpf et. al. (2021) [1]. The paper is very careful in acknowledging left-lateralization (see Page 2 of the Schrimpf paper).
    * The introduction should present some more details on the setting. What language models are used? What data? This is covered later but is missing from an introduction that should introduce more of the paper. Also the introduction doesn’t even mention the other contribution of the paper on model size correlating with neural fit.
* The novelty and takeaways of this paper are unclear to me.
    * First, the paper claims that larger language models have better fits. Schrimpf et. al. (2021) [1] already establishes that. This is acknowledged in the conclusion/discussion section. The authors also refer to Antonello et. al. 2024 which seems to establish this as well. How would the authors characterize the difference in the takeaway of their paper vs the prior takeaway? I’m not particularly satisfied with the argument that much larger models were used or that a logarithmic scaling law was identified.
    * Caucheteux and King [2] discuss lateralization at length in their paper. Refer to page 3 where they show that language embeddings have significantly better left-lateralization over right-lateralization (R±0.01, p < 10^-14). This may not use the controls presented in this paper but still points to the same result.
    * I think both of these need to be discussed at more length. The contribution of this paper should be better characterized. I don’t see the novelty of this work even in regards to lateralization.
* Baselines
    * The paper doesn’t consider the baseline comparison with randomly initialized models. Why? I think this is a very important baseline for characterizing architectural bias and this is done in previous work.

[1] Schrimpf et. al. The neural architecture of language: Intergrative modeling converges on predictive processing. PNAS 2021.
[2] Caucheteux et. al. Brains and algorithms partially converge in natural language processing. Nature communications, 2022.

**Questions:**

* Why did you also include Mamba? I’m not sure it adds anything or detracts anything but I’m just curious about the choice.

**Limitations:**

I believe all reasonable limitations were addressed with regards to this paper.

---

> ### Author Rebuttal · Authors · 2024-08-05
>
> Thank you for your feedback. Here is a point-by-point response to your comments.
>
> > The paper should make an argument about why lateralization is an important property of brain fits with LLMs. I think this was missing.
>
> Studies correlating word embeddings or LLMs activations and fMRI data have produced strikingly bilateral results. One potential explanation is that brain scores are essentially driven by semantic representations which are supposedly represented in a very distributed fashion across both hemispheres. Our work shows that the symmetrical results are due to the use of models with small numbers of parameters and that when the number of parameters increases, the left-right asymmetry becomes larger and larger. This non obvious empirical observation opens several questions for further investigations, notably what aspects of large LLM relative to small ones make them more asymmetrical.
>
> > A small nitpick: From my understanding, I can’t see any data from right hemisphere regions in Schrimpf et. al. (2021) [1]. The paper is very careful in acknowledging left-lateralization (see Page 2 of the Schrimpf paper).
>
> Schrimpf et al (2021) show results from both hemispheres in Fig. S3 of their supplementary materials. Responses on both sides are remarkably similar and the caption states that "The distribution of predictivity values across the language-responsive voxels [...] are similar across regions, and between the LH and RH components of the network". The authors do not report any quantitative comparison between hemispheres. While they acknowledge the well-known fact that language is left lateralized, surprisingly, they did not discuss the lack of lateralization in their results. Our work replicates their findings with small models but shows that it breaks down with larger models.
>
> > The introduction should present some more details on the setting. What language models are used? What data? This is covered later but is missing from an introduction that should introduce more of the paper.
>
> The introduction only contained sparse information about the exact list of language models that are used, but all such information is provided in great details both in the Methods section (that follows the Introduction) and the Appendices. As the final camera-ready version of the paper allows for one extra page, we will be able to present more details as suggested.
>
> > First, the paper claims that larger language models have better fits. Schrimpf et. al. (2021) [1] already establishes that. This is acknowledged in the conclusion/discussion section. The authors also refer to Antonello et. al. 2024 which seems to establish this as well. How would the authors characterize the difference in the takeaway of their paper vs the prior takeaway?
>
> Our discussion simply mentions that we replicate these previous findings and devotes more space on the growing left-right asymmetry which is a genuine novelty.
>
> > Caucheteux and King [2] discuss lateralization at length in their paper. Refer to page 3 where they show that language embeddings have significantly better left-lateralization over right-lateralization (R±0.01, p < 10^-14). This may not use the controls presented in this paper but still points to the same result.
>
> Thank you for pointing to the left-right test by Caucheteux & King (2022), which we had missed. Note that although they test it, they do not really discuss nor comment on it in their paper.
>
> Nevertheless, we believe that our paper goes well beyond showing a left-right difference in a single model. Given one model, if the data has higher signal-to-noise ratio in the LH, one expects an encoding model to better fit the LH compared to the RH. This is in fact the case in the Caucheteux & King (2022; Fig. 2d).
>
> It is a legitimate concern that if the signal-to-noise is higher in the left hemisphere, then the brain correlations, as predicted by the encoding models, should also be higher in general. But, given that the data is constant across models, it is unclear why this would affect the slope in comparing brain correlations and number of parameters. We did some extra analyses (see pdf)  showing that, although the mean brain correlation depends on signal-to-noise ratio, the slope cannot be explained by this factor.
>
> > The paper doesn’t consider the baseline comparison with randomly initialized models. Why? I think this is a very important baseline for characterizing architectural bias and this is done in previous work
>
> Fig. 2a of the submission presents the distributions of r-scores from various models, including distributions from random (fixed) embedding models. Pasquiou et al (2022, ICML) have shown that this type of model yields brain correlations which are as strong or stronger than untrained language models. Nevertheless, following the reviewer’s comment, we computed the brain scores for untrained versions of the four variants of GPT-2. We replicate Pasquiou et al.’s observation, that is, all four untrained models performed below the 1024d random baseline which achieved an average brain score of 0.182 (on the 25% most reliable voxels). For example, gpt2-medium, which has the same number of dimensions, yields a brain correlation of 0.168 (vs. 0.417 for the corresponding trained model). While the left-right score difference increases with the number of parameters (r=0.99, p=8.3e-3), the relationship breaks down and is flat in the case of the untrained models (r=-0.03, p=0.97).
>
> > Why did you also include Mamba? I’m not sure it adds anything or detracts anything but I’m just curious about the choice.
>
> Mainly out of curiosity. It is the first time that a large language model not based on the Transformer architecture has competitive performance on par with more traditional Transformer-based language models. As discussed in the Results section, the Mamba family has similar performance as encoding models of brain functional data, which was not a priori obvious.

---

> ### Comment · Reviewer_Wj1c · 2024-08-10
>
> Thank you for the detailed response and clarification. This was very useful for me to understand the positioning of this paper. Although this may be my own personal opinion, I think it would be great to characterize this contribution through the lense of what you wrote in this response. I also really appreciate your response on why lateralization was important! My main feedback would be to provide some more framing on the connection of increased lateralization with increasingly large language models.
>
> In general, I also really like the randomly initialized result. This gives me confidence in conclusions drawn in the paper. I understand that prior papers may have consistently shown the same result but I would strongly push for this baseline to properly characterize the role of linguistic learning in a camera ready version of the paper.
>
> In light of the response, I will raise my score to reflect my better understanding of the contribution.

---

> > ### Author Response · Authors · 2024-08-12
> >
> > Thank you for your constructive feedback. We will try to better clarify our position in the camera-ready paper.
> >
> > > I would strongly push for this baseline to properly characterize the role of linguistic learning
> >
> > We will include these analyses in the final version of the paper.
> >
> > Our interpretation for the fact that baselines with untrained contextual models perform less well than fixed random embeddings is the following. The linear regression to fit the encoding models to brain data may learn some association between the (random) embedding associated with a given word and the brain activation elicited by this word. Contextual models (transformers, RNN, …) would then perform less well than fixed embedding models as the context introduces noise in the activation pattern associated with a given word.

---

### Author Rebuttal · Authors · 2024-08-05

We would like to sincerely thank all four reviewers for their detailed and valuable reviews of our manuscript. We think they have helped to clarify our contribution and strengthen our paper by adding some checks, as described below.

An important point, raised by most reviewers, concerns the potential impact of differences in signal-to-noise ratio in the right and left hemispheres. Indeed, in regions where the signal-to-noise ratio is stronger, one intuitively expects higher correlations between brain data and model predictions. Could this explain away our findings?

To address this issue, we conducted new analyses and generated figures in the accompanying pdf.
Fig.1 provides more details about inter-subjects correlations (model free) in the left and right hemispheres. In a nutshell, the inter-subjects correlation (ISC) follows the diagonal in left vs. right plot as a general trend, but several regions in the left hemisphere do indeed show higher reliability.
Fig. 2 shows that in the different regions of interest the average brain correlation follows the reliability of the signal (ISC), but the slope in the relationship between brain correlation and number of parameters do not generally follow such a trend.
So, differences in signal-to-noise ratio partly explain the mean brain score difference between left and right, but do not explain the increase in the difference. In other words, while a signal-to-noise ratio difference may explain an asymmetry in a single model, it does not account for the fact that larger models fit better and better left hemispheric activations than right hemispheric ones.

In addition, since the original submission, we have had time to extend the work to Chinese and French data from the Le Petit Prince dataset. Even though we found fewer pretrained large language models available for these two target languages, the analyses essentially reproduce the results obtained with English, in particular with respect to the scaling law in the left-right difference in brain correlation. We intend to add these additional results to the camera-ready paper.

---

### Decision · Program_Chairs · 2024-09-25

**Decision:**

Accept (poster)

**Comment:**

This paper presents a novel result about the asymmetry of language representations in natural speech understanding when modeled with language model of increasing complexity. This is an important question because there is a disagreement in the field: traditional analyses always focus on the left hemisphere, while experiments with natural stimuli often report bilateral representations. The authors proceed by testing models of various sizes and replicating the existence of a scaling law relating the number of model parameters and the ability to predict fMRI activity. They observe that this pattern exhibits a significant Left-Right asymmetry. The reviewers have appreciated the strength of the arguments, especially after adding the additional analysis. The recommendation is to accept this paper as long as the author incorporate the new analyses in the manuscript, along with the appropriate discussion. Is it also particularly valuable to expand on the potential theoretical reasons for this result, since while it does agree with the historical literature on language processing, it is different from what has been recently reported in studies using language models.